# Perinatal and Demographic Risk Factors Associated with Autism Spectrum Disorder: A National Survey of Potential Predictors and Severity

**DOI:** 10.3390/healthcare12202057

**Published:** 2024-10-16

**Authors:** Aikaterini Sousamli, Elena Dragioti, Dimitra Metallinou, Aikaterini Lykeridou, Panagiota Dourou, Chrysoula Rozalia Athanasiadou, Dimitrios Anagnostopoulos, Antigoni Sarantaki

**Affiliations:** 1Midwifery Department, Faculty of Health and Care Sciences, University of West Attica, 12243 Athens, Greece; dmetallinou@uniwa.gr (D.M.); pdourou@uniwa.gr (P.D.); esarantaki@uniwa.gr (A.S.); 2Research Laboratory Psychology of Patients, Families, and Health Professionals, Department of Nursing, School of Health Sciences, University of Ioannina, 45500 Ioannina, Greece; dragioti@uoi.gr; 3Section of Social Medicine-Psychiatry-Neurology, School of Medicine, National and Kapodistrian University of Athens, 10559 Athens, Greece; danagnostopoulos@uoa.gr; 4Research Lab (PEARL)—Perinatal Care and Counseling for Special Populations, Department of Midwifery, University of West Attica, 12243 Athens, Greece

**Keywords:** autistic spectrum disorders, functioning, prenatal risk factors, maternal age, age of diagnosis, vaginal bleeding, prenatal infections, family income, birth order, family history

## Abstract

**INTRODUCTION:** This study investigates autism spectrum disorders (ASD) in Greece, focusing on estimating prevalence and identifying regional disparities in children aged 4 to 7 years. **MATERIALS AND METHODS:** Utilizing a quantitative, descriptive, and exploratory methodology, the research employed a structured questionnaire to gather extensive maternal and child health data. **RESULTS:** The sample consisted of 517 mothers of children diagnosed with ASD from all over Greece, contributing to a nuanced understanding of ASD predictors. This study aims to elucidate the role of prenatal factors in the likelihood of an ASD diagnosis and their impact on the subsequent functionality of children with ASD. The study identified significant predictors of lower functionality in children with ASD, including higher maternal age, delayed ASD diagnosis, lower family income, and higher birth order. Prenatal health issues, such as vaginal bleeding and infections, also influenced functional outcomes. Notably, a family history of neurological or psychiatric conditions appeared protective. **DISCUSSION:** The regression model demonstrated robust predictive power, underscoring the complexity of genetic, environmental, and socioeconomic factors in ASD development. The findings advocate for early diagnosis and intervention, systematic screening, and addressing socioeconomic disparities to improve functional outcomes. The results support evidence-based service development and policy adjustments to enhance early identification, intervention, and rehabilitation for children with ASD. **CONCLUSIONS:** Establishing standardized case-recording procedures and an ASD register at national and regional levels is recommended for systematic monitoring and resource evaluation.

## 1. Introduction

Autism spectrum disorders (ASD), evident from early childhood, are neurodevelopmental conditions characterized by deficits in communication, socialization, and cognitive abilities. Common features include restricted interests, repetitive behaviors, and challenges in social interactions and relationships [1]. Globally, ASD diagnoses in children are increasing, with the U.S. showing a significant rise to 1 in 36 children [2]. ASD is four times more prevalent in boys than girls, potentially due to male fetuses’ greater vulnerability to prenatal stressors [3,4,5]. Causes may involve defective genes, chromosomal abnormalities, medical conditions, viruses, prenatal stressors, and environmental factors affecting brain development and central nervous system function [1,2,3,4,5,6]. Early diagnosis and intervention (birth to 36 months) significantly enhance a child’s development [7]. Diagnosis relies on clinical criteria from the International Classification of Diseases (ICD-10) and the Diagnostic and Statistical Manual of Mental Disorders (DSM-5) [8,9]. Symptom severity varies from mild to severe, affecting the level of support needed for social communication and repetitive behaviors [10].

Families of children with ASD may exhibit characteristics that are correlated with various factors such as advanced parental age [11,12,13,14], low socioeconomic status [5,15], and psychiatric history [16]. Other neurodevelopmental disorders in siblings [17], infertility history [3], maternal exposure to pesticides, and toxic chemicals [18,19], as well as loud noises during pregnancy [3], have also been associated with ASD. Additionally, maternal behaviors and conditions such as smoking, alcohol consumption [20], experiencing obstetric complications [3,12], hyperemesis, epilepsy, hypertension, and polycystic ovary syndrome [21] show correlations with ASD diagnoses. The use of fish oil supplementation during pregnancy [22], birth order [11,13], and family size have similarly been identified as factors associated with ASD [23]. Furthermore, conditions like preeclampsia, eclampsia [24], iron deficiency [25], and immune stress from maternal inflammation and third-trimester infections like rubella, cytomegalovirus, or Toxoplasma gondii are linked to ASD risk [1,6,21].

Pre-pregnancy overweight, particularly with rapid weight gain during pregnancy, is identified as a risk factor for gestational diabetes [12,13]. This condition is linked to neurodevelopmental disorders in offspring [26].

Research indicates that ASD is frequently linked to multiple gestation [13], maternal medical interventions during labor, such as oxytocin exposure with epidural analgesia [27], preterm delivery [12,28], and various pregnancy and delivery complications [21], including cesarean section [12], abnormal fetal presentation, fetal distress, postpartum hemorrhage due to uterine atony, and prolonged labor [13]. Newborn studies highlight issues such as low Apgar scores [13], oxytocin-induced labor, low birth weight [12,13,14,21], birth asphyxia, infections, epilepsy [21], and neonatal complications [13,21].

The severity of intellectual disability is determined by the level of adaptive functioning impairment, emphasizing the importance of both IQ and adaptive functioning in the assessment of ASD. Researchers investigated the relationship between intrauterine toxicity and labor complications with ASD symptoms, IQ, and adaptive functioning. [28].

Recent epidemiological findings [29,30] provide a clearer picture of ASD prevalence within the Greek context. Kouznetsov et al., in their 2023 study [29], estimate the prevalence among 2–17-year-olds at 0.94%, demonstrating a significant gender disparity (1.44% in males versus 0.41% in females; *p* < 0.001). In the year 2021, there were 15,706 ASD diagnoses in this age group, with 12,380 boys and 3325 girls affected. These findings highlight the urgent need for research and interventions tailored to Greece’s demographic and regional specifics regarding ASD.

This study aims to identify the prenatal, perinatal, and postnatal factors associated with ASD diagnoses in Greece. This involves examining how specific environmental and genetic factors correlate with ASD incidence. Few reports have addressed prenatal influences on the severity of clinical presentations in children with ASD, particularly regarding their adaptive and intellectual functioning, especially in Greece [31]. Factors such as deficient birth weight (<1500 g), premature delivery, and low Apgar scores may directly affect adaptive functioning, while gestational diabetes and advanced parental age were not highlighted [14,31]. Hadjkacem et al. [32] found no significant association between clinical severity and prenatal factors in children with ASD. Conversely, another study indicated that preeclampsia and gestational diabetes are associated with greater overall severity, especially in stereotypical behavior and sociocommunicative impairments in children with ASD [33].

Our research aims to enhance the existing literature and address research gaps by identifying and analyzing correlations between various study variables. It seeks to provide prenatal data on ASD in Greece to support evidence-based planning and resource development nationally and regionally. Specifically, the objectives include quantifying the influence of prenatal exposure to environmental toxins on the risk of ASD diagnosis, assessing the impact of familial genetic history on ASD incidence, and evaluating how these factors vary regionally across Greece. This analysis aims to contribute to targeted public health strategies and inform policy development for early ASD detection and intervention. The research questions for this study are as follows: (1) What are the characteristics associated with varying levels of functionality in children with ASD in a Greek sample? (2) How do sociodemographic, clinical, and perinatal factors compare concerning functionality levels in Greek children with ASD? (3) Which factors serve as the most robust predictors of functionality in our study participants with ASD?

The prevalence of ASD has significantly risen over the past 20 years [34], posing serious social challenges due to population growth and escalating rehabilitation costs. Our findings are likely to support the theory that ASD is multifactorial, with certain risk factors demonstrating statistical significance. This research was conducted to examine whether different prenatal factors, obtained from maternal self-questionnaires, are correlated with increased severity in the clinical presentation of ASD, as well as with intellectual and adaptive functioning, in a group of children who were diagnosed with ASD and underwent comprehensive clinical evaluations.

## 2. Materials and Methods

In this study, a correlational research design was employed to investigate the associations between perinatal exposures and ASD outcomes across various regions of Greece. It does not manipulate variables but observes existing conditions and their interrelations. Purposive sampling was employed to select a specific cohort of participants—mothers of children diagnosed with ASD born between 2017 and 2020. The sampling process ensured that participants were drawn from diverse geographic locations and socioeconomic backgrounds, representative of the broader Greek population.

To collect data, a structured questionnaire was developed (Appendix A), comprising both closed-ended and open-ended questions. This questionnaire encompassed demographic, clinical, and prenatal/perinatal health information, with an emphasis on factors potentially associated with ASD. The structured questionnaire used in this study collected data on a variety of variables, including maternal demographics (age, education, family income), pregnancy and delivery characteristics (gestational age, birth complications), and child characteristics (age at diagnosis, functionality level of ASD). The questionnaire was validated in a pilot study prior to distribution. Only mothers participated in the study, as they served as the primary caregivers for their children and had extensive knowledge of their pregnancy, delivery, and postpartum experiences. This targeted methodological approach was essential for focusing the research on a demographic that could provide the most pertinent and comprehensive information regarding prenatal, perinatal, and early childhood factors associated with ASD diagnoses.

The questionnaires were disseminated to specialized educational institutions, day centers, Pediatric and Developmental Health Clinics, and Disabled and Special Needs Children’s Clubs across Greece. To minimize potential biases, the research team ensured high participation rates by contacting centers twice during the data collection phase. Data stratification was implemented based on geographic and socioeconomic variables, ensuring a proportional representation of urban and rural regions. Furthermore, weighted analysis techniques were employed to adjust for any disparities in sample representation across strata, thereby enhancing the generalizability of the findings to the national population. The sample comprised 517 mothers, representing a diverse cross-section of the Greek population.

A telephone call was made, and an email was sent to representatives of the aforementioned institutions (the school director, the scientific manager of the day center and developmental health clinic, or the president of the Special Needs Children’s Clubs) in 340 Greek centers to explain the study’s objectives and goals and to facilitate further communication. The research team contacted each center at least twice during the questionnaire collection period to ensure and minimize errors and biases in the collection process. These centers provide support and counseling to children and their families, intending to promote inclusive education and ensure the optimal integration of every child into the appropriate educational setting. Participants who did not possess adequate proficiency in the Greek language were not included in the study.

The study’s sample was proportionally stratified to ensure a representative distribution across the country. Specifically, of the 517 mothers included, 20% (103) were from Central Macedonia and Thrace, 30% (155) resided in Epirus and Thessaly, 15% (78) were located in Crete, and the remaining 35% (181) were distributed across Attica, Peloponnese, and the Aegean islands. Furthermore, 40% of the sample was recruited from Specialized Educational Institutions (e.g., special needs schools and autism-specific day schools), 30% from Therapeutic Day Centers, 20% from Community-based Support Centers, and the remaining 10% were recruited from Pediatric and Developmental Health Clinics. The sampling strategy was designed to reflect the demographic distribution of the Greek population. The sample was stratified based on key geographic and demographic variables to ensure comprehensive representation. These strata encompassed various regions of Greece, urban versus rural settings, and socioeconomic backgrounds. The study ensured that the sample included participants from all administrative regions of Greece, proportionate to the population size of children aged 4–7 years in each region, as reported by the latest national census data.

Additional stratification was applied based on socioeconomic status, wherein participants were grouped into quartiles based on the income distribution data available from national statistics. The income quartiles were delineated as follows: Quartile 1: less than EUR 10,000; Quartile 2: EUR 10,001 to EUR 20,000; Quartile 3: EUR 20,001 to EUR 40,000; and Quartile 4: more than EUR 40,001. These ranges reflect the national income distribution as per the most recent data available from the Hellenic Statistical Authority (accessed on 12 October 2024), which can be found at (https://www.statistics.gr/documents/20181/262f2183-1930-6ece-a792-e02103dfbe00). This stratification facilitated the analysis of the impact of socioeconomic factors on the prevalence and characteristics of ASD among children. In our analysis, we employed statistical techniques appropriate for stratified sampling data, including weighted analysis to account for potential disparities in sample representation across different strata. This methodological approach enhances the generalizability of our findings to the national population.

The researchers of the present study conducted a correlational analysis using data collected from health records and structured surveys distributed across multiple healthcare centers in Greece. This approach allows for the examination of relationships between observed variables without experimental manipulation. They aimed to achieve accurate and scientifically reliable outcomes by concentrating on pivotal aspects, including the creation of an appropriate questionnaire and employing a proportionate sampling design for the Greek population. Specifically, for the study’s application within the Greek population, the functional status of the children was assessed using three distinct categories of functionality (high, moderate, and low).

The questionnaire referred to children diagnosed with “Autism spectrum disorders” according to DSM-5 or “Autism” as defined by ICD-10. A child can only be classified as “ASD” based on a clinical diagnosis issued by a public sector child neurologist, child psychiatrist, or developmental pediatrician.

The survey questionnaires were distributed via postal mail, with dissemination to remote areas of Greece conducted over a three-month period at the expense of the research team. The primary objective was to protect the personal data of the individuals included in the sample. To this end, each participant was informed of the strict confidentiality of their personal information, and the research team assured them that the data would be utilized solely for academic purposes. After completing the distribution process, the leading researcher collected the completed questionnaires via mail and prepaid courier service. With the understanding that they concurred with the aforementioned, participants signed the consent form, and the process continued. The participants in the study provided their informed consent to participate voluntarily and confidentially, as indicated by their completion of the questionnaire. This consent was obtained following a thorough explanation of the study’s purpose, nature, and data handling procedures.

Only members of the research team had access to the data, which were utilized solely for scientific purposes in alignment with the study’s objectives. Before signing the consent form, participants were informed both verbally and in writing about the study’s aim, confidentiality, anonymity, voluntary participation, and the option to withdraw at any time. The protection of the participants’ personal data was ensured through the anonymous completion of the questionnaires.

Before disseminating the questionnaires, a robust coding process was implemented to safeguard the anonymity of all individuals involved. An exclusive identifier code was allotted to each participant, which served to replace any personal information on the questionnaire with the assigned code. This code was employed to track the responses without revealing the identity of the respondents. The coding system was designed to ensure that only the principal researcher and the supervisor of the study had access to the key that linked the codes to specific individuals, and this key was stored securely in an encrypted digital format. Additionally, all data collected were entered and analyzed anonymously, with findings reported in aggregate form to further prevent the identification of any participant. These measures were crucial to maintaining the trust of the participants and the integrity of the research process. During the course of the study, participants were provided with either face-to-face or telephone support to assist them in completing the questionnaires, should they encounter any difficulties. The team’s cohesion and cooperation facilitated the resolution of any consultation or persuasion difficulties with the groups.

This research was undertaken upon obtaining approval from the Ethics Committee of the University of West Attica, located in Athens (Reference Number: 29346/08-04-2024). The data collection phase extended from April 2024 to June 2024.

Descriptive statistics were calculated to summarize the characteristics of the study sample. This included calculating means, standard deviations (SDs), medians, and ranges for continuous variables such as age, weight, height, gestational age, length of labor, and birth weight. For categorical variables (i.e., sociodemographic characteristics, ASD-related and peri- and postnatal variables), frequencies and percentages were determined to describe the distribution of family structure, education level, family income, geographic residence, functional level of children with ASD, gender distribution, birth order, and family history of developmental or psychiatric disorders.

In the collection of data regarding familial health history and comorbid conditions, participants provided information through self-administered questionnaires. Parents were asked to report any known diagnoses of autism, developmental disorders, epileptic seizures, depression, or anxiety disorders within the family. It is important to note that these data were self-reported and not independently verified against medical records due to the logistical constraints of this study’s broad geographic scope. Regarding the analysis of comorbidities, the initial design of our data collection instruments did not include detailed demographic breakdowns such as the division of comorbidities by gender. As such, while our dataset provides a general overview of comorbid conditions reported by participants, it does not allow for a reliable disaggregation of these conditions by gender.

To compare the characteristics of children with high versus low/moderate functionality in ASD, inferential statistical tests, including independent sample *t*-tests and Chi-square tests, were performed. In the current analysis, the dependent variable was the level of functionality in children with ASD, categorized as high versus low/moderate functionality. The independent variables covered sociodemographic characteristics, ASD-related factors, as well as peri- and postnatal variables. In the categorization of functionality levels among children with ASD, middle- and low-functioning levels were merged based on preliminary findings that indicated substantial overlap in their developmental and behavioral profiles. This amalgamation was intended to enhance the statistical robustness of our analyses by reducing variability that does not contribute to distinct differences in outcomes. A detailed justification for this decision is provided in the Results section, where we discuss how this approach impacts our findings. Furthermore, a binomial logistic regression model was utilized to identify predictors of functionality in children with ASD, using the previously mentioned sociodemographic, ASD-related, and peri- and postnatal variables as independent variables.

In this analysis, we included only the significant variables derived from inferential statistical tests to further account for their potential influence on the functionality outcomes. Odds ratios (OR) and 95% confidence intervals (CI) were reported. The model’s fit was evaluated using the Akaike information criterion (AIC) and pseudo-R^2^ values (Cox and Snell’s R^2^ and Nagelkerke’s R^2^). Multicollinearity was assessed using variance inflation factor (VIF) and tolerance. Tolerance values below 0.1 (equivalent to VIF above 10) indicate serious multicollinearity. The model’s predictive accuracy, sensitivity, specificity, and area under the curve (AUC) were also assessed to ensure its reliability and discriminatory ability. All analyses were performed in Jamovi (version 2.5.5), and the visualizations were created using R (version 4.4.1). Statistical significance was set at *p* ≤ 0.05.

## 3. Results

### 3.1. Descriptive Statistics of Continuous Variables

Appendix A presents the descriptive statistics for a range of continuous variables pertinent to the study. The mean age of mothers in the sample was 40.2 years (SD = 5.42), with a median age of 40 years, and ranging from 21 to 63 years. Fathers were, on average, 43.3 years old (SD = 6.19), with a median age of 43 years, and ages ranging from 24 to 70 years. The mean age of children diagnosed with ASD was 6.89 years (SD = 3.94), with a median of 6 years and an age range of 1 to 28 years. The average age at ASD diagnosis was 3.47 years (SD = 2.29), with a median age of 3 years, ranging from 0 to 24 years.

Pre-pregnancy weight of mothers had a mean of 67.2 kg (SD = 33.4), a median of 63 kg, and ranged from 34 to 76.1 kg. At the end of pregnancy, the mean weight was 80.4 kg (SD = 29.1), with a median of 78 kg, and ranged from 49 to 65.1 kg. The mean height at the end of pregnancy was 164 cm (SD = 16.1), with a median of 165 cm, and a range from 1.52 to 188 cm. Fathers’ age at conception averaged 36.0 years (SD = 5.61), with a median of 36 years, and ranged from 20 to 65 years. Gestational age at birth had a mean of 37.9 weeks (SD = 2.74), with a median of 38 weeks, and ranged from 9 to 42 weeks. The mean duration of labor was 4.88 h (SD = 6.14), with a median of 2 h, and ranged from 0 to 48 h. The mean birth weight of children was 3089 g (SD = 744), with a median of 3150 g and a range of 2900 to 4850 g.

### 3.2. Descriptive Statistics of Sociodemographic Variables

Appendix A summarizes the sociodemographic characteristics of the sample. Most families consisted of two parents living together (90.1%), with a smaller proportion of single-parent families (4.6%). Regarding educational attainment, 43.5% of mothers had tertiary education, followed by 28.4% holding a master’s degree. Fathers’ education levels showed 47.6% had secondary education, while 35.0% had tertiary education. Annual family income predominantly fell within the EUR 20,001–40,000 range (42.1%), followed by EUR 10,001–20,000 (38.1%). The majority of families resided in Attica (36.6%) and Central Macedonia (13.4%), with other regions having smaller representations.

### 3.3. Descriptive Statistics of ASD-Related Variables

The majority of children with ASD (62.3%) were classified as having high functionality, while 29.4% exhibited moderate functionality and 8.3% demonstrated low functionality. To ensure sufficient statistical power for group comparisons, the low and moderate functionality categories were combined into a single group (low/moderate functionality). As detailed in Appendix A, the final distribution of functionality levels was 62.3% high and 37.7% low/moderate. The functionality levels of children with ASD were categorized as high (62.3%) and low/moderate (37.7%).

A substantial majority of the children with ASD were male (81.0%), with females constituting 19.0%. The majority of these children were firstborn (67.3%), followed by second-born children (26.5%). Only 10.6% of families reported having additional children with neurodevelopmental difficulties. Analysis of family history data indicated that 32.3% of mothers and 39.1% of fathers had a documented history of autism, developmental disorders, epileptic seizures, or depression/anxiety disorders, based on self-reported information.

### 3.4. Descriptive Statistics of Peri- and Postnatal Variables

Appendix A provides an overview of peri- and postnatal variables. Exposure to chemicals during pregnancy was reported by 6.0% of mothers, while 19.0% smoked during pregnancy. Gestational diabetes was present in 18.8% of cases, and 13.2% of mothers experienced hyperemesis. Regarding delivery, 62.9% of births were via cesarean section, while 37.1% were normal deliveries. Exclusive breastfeeding was reported for 43.3% of newborns, whereas 47.0% received mixed feeding (breastfeeding and formula). Additionally, 22.6% of newborns were exclusively formula-fed.

### 3.5. Comparison of Continuous Variables by Functionality of Child with ASD

Appendix A provide the results of independent sample t-tests and group statistics comparing various continuous variables based on the functionality of children with ASD (categorized as low/moderate and high). There was a significant difference in the age of mothers between the two groups (t(515) = 2.214, *p* = 0.027). Mothers of children with low/moderate functionality were older (M = 40.86, SD = 5.38) compared to mothers of children with high functionality (M = 39.78, SD = 5.41). A highly significant difference was found in the age at ASD diagnosis (t(515) = −5.431, *p* < 0.001). Children with high functionality were diagnosed later (M = 3.89, SD = 2.59) compared to those with low/moderate functionality (M = 2.78, SD = 1.46). There was also a significant difference in the age of mothers at conception between the two groups (t(515) = 2.443, *p* = 0.015). Mothers of children with low/moderate functionality were older at conception (M = 33.63, SD = 5.05) than mothers of children with high functionality (M = 32.53, SD = 4.86).

Figure 1 illustrates the distribution of various demographic and medical variables between two groups of children with ASD: those with low/moderate functionality (blue) and those with high functionality (red). Key variables are plotted on the *x*-axis, including the mother’s age, father’s age, age of the child with ASD at diagnosis, the pre-pregnancy weight of the mother, weight at conception, weight at end of pregnancy, gestational age at birth, duration of labor, and birth weight of the child. The *y*-axis represents the means of these variables, expressed in relevant units (years, kilograms, weeks, hours, grams). Each box plot displays the interquartile range (middle 50% of data), with lines extending to the minimum and maximum values within 1.5 times the interquartile range from the box. Means are marked by diamonds, providing a clear reference for direct comparisons between the two groups. This visualization aids in identifying potential patterns or significant differences in demographic and medical variables between children with varying levels of ASD functionality.

Continuous variables, such as the father’s age, age of the child with ASD, pre-pregnancy weight, weight at the end of pregnancy, height at the end of pregnancy, father’s age at conception, gestational age at birth, duration of labour, and birth weight of the child, did not show significant differences between the two groups (*p* > 0.05).

### 3.6. Comparison of Sociodemographic Categorical Variables by Functionality of Child with ASD

The analysis of sociodemographic categorical variables (Appendix A) using the Chi-square test showed that there were significant differences only in annual family income between the groups (χ^2^ = 12, df = 4, *p* = 0.017). Families of children with low/moderate functionality were more likely to fall into lower income brackets compared to families of children with high functionality, as depicted in Figure 2. The other sociodemographic categorical variables (family type, mother’s educational level, and father’s educational level) did not show significant results, as their *p*-values were greater than 0.05 (Figure 2). This was also the case for the current family residence (Appendix A).

### 3.7. Comparison of ASD-Related Categorical Variables by Functionality of Child with ASD

The analysis of ASD-related categorical variables (Appendix A) using the Chi-square test showed that there were significant differences in the birth order of the child and their functionality with ASD (χ^2^ = 33.3, df = 5, *p* < 0.001), as well as in the mother’s family history with autism, developmental disorders, epileptic seizures, depression, or anxiety disorder (χ^2^ = 12.2, df = 1, *p* < 0.001). Specifically, the first-born children and second-born children had a higher proportion of high functionality compared to later-born children (third child, fourth child, etc.), and a higher proportion of children with a high level of functionality had a mother’s family history of these conditions compared to children with a low/moderate level of functionality (Figure 3).

### 3.8. Comparison of Peri- and Postnatal Categorical Variables by Functionality of Child with ASD

The analysis of peri- and postnatal categorical variables (Appendix A) using the Chi-square test showed significant differences in several factors associated with the functionality of children with ASD. These factors include hyperemesis (severe vomiting) (χ^2^ = 5.3, df = 1, *p* = 0.03), viral or bacterial infection (χ^2^ = 3.87, df = 1, *p* = 0.05), vaginal bleeding (χ^2^ = 4.28, df = 1, *p* = 0.04), vaginal bleeding during the pregnancy of the child with ASD (χ^2^ = 12.6, df = 1, *p* = 0.006), and whether the baby cried immediately after birth (χ^2^ = 15, df = 1, *p* = 0.001). Specifically, severe vomiting during pregnancy, viral or bacterial infections during pregnancy, vaginal bleeding during pregnancy, vaginal bleeding during the pregnancy of child with ASD, and the baby crying immediately after birth are associated with lower functionality in children with ASD (Figure 4).

### 3.9. Multivariate Analysis of Factors Related to the Functionality of Child with ASD

The binomial logistic regression model (Table 1) revealed significant predictors of functionality in children with ASD. For each additional year in the maternal age, the probability of the child exhibiting low or moderate functionality increases by 6% (OR = 1.06, 95% CI [1.00, 1.12]). Although statistically significant, this effect is relatively modest. Mothers of advanced maternal age (≥40) have a higher incidence of severe autism cases compared to younger mothers (117 vs. 78).

Conversely, for each additional year’s delay in diagnosing ASD, the probability of the child displaying low or moderate functionality decreases by 35% (OR = 0.65, 95% CI [0.55, 0.77]), suggesting that those diagnosed later tend to have milder symptoms (105 with high functionality vs. 59 with low to moderate functionality).

Children from families earning below 10,000 EUR are 2.58 times more likely to exhibit low or moderate functionality compared to those from families earning between 10,001–20,000 EUR (OR = 2.58, 95% CI [1.21, 5.50]), demonstrating the substantial influence of lower socioeconomic status on child development. In this study, among 44 children from families earning below 10,000 EUR, 26 (or 59%) present with severe autism (low or moderate functionality).

Second-born children exhibit a 2.71 times higher likelihood of demonstrating low or moderate functionality compared to first-born children (OR = 2.71, 95% CI [1.70, 4.29]). Specifically, the findings indicate that among first-born children, approximately 29.6% (103 out of 348) present with severe autism, whereas among second-born children, approximately 56.9% (78 out of 137) manifest severe autism. These results suggest that birth order exerts a significant influence on the child’s developmental outcomes.

Children whose mothers have a family history of autism, developmental disorders, epileptic seizures, depression, or anxiety are 42.5% more likely to have low or moderate functionality (severe autism) compared to high functionality (mild autism) (OR = 0.58, 95% CI [0.36, 0.91]). This finding suggests that such a familial history may be associated with a reduced probability of higher functionality in individuals with ASD. However, the univariate analysis revealed that children with a familial history demonstrate a higher prevalence of mild autism (73.05%) compared to children without a familial history (57.14%). This observation appears to contradict the regression findings, which account for multiple variables.

Mothers who experienced a viral or bacterial infection during pregnancy were 51% less likely to have children with high functionality (OR = 0.49, 95% CI [0.25, 0.97]). The findings indicate that 73.4% (47 out of 64) of children whose mothers had an infection exhibited severe autism (low to moderate functionality). This observation emphasizes the potential adverse effects of maternal infections on the child’s developmental outcomes.

Vaginal bleeding during pregnancy was associated with significant risks: mothers who experienced bleeding during the first trimester exhibited a 6.43-fold increased likelihood of having children with low or moderate functionality (OR = 6.43, 95% CI [1.55, 26.77]), while third-trimester bleeding increased this likelihood by 13-fold (OR = 13.02, 95% CI [2.15, 78.93]). The study revealed that 46.3% of children whose mothers experienced vaginal bleeding during pregnancy demonstrated severe autism, compared to 35.4% for mothers who did not experience bleeding, while 68.42% (13 out of 19) of ASD children with third trimester bleeding presented with severe autism. These findings underscore the potential risks that pregnancy complications may pose to child development.

Finally, infants who exhibited immediate post-natal crying were 61% less likely to present with low or moderate functionality (OR = 0.39, 95% CI [0.19, 0.81]). Among neonates who demonstrated immediate post-natal crying, 33.96% (144 out of 424) were diagnosed with severe autism. This early health indicator suggests a strong positive association with enhanced functionality.

The model demonstrated a good fit, with several measures indicating its adequacy. The deviance of the model was 554, and the AIC was 598. The pseudo-R^2^ values, including Cox and Snell’s R^2^ (0.220) and Nagelkerke’s R^2^ (0.300), suggested a reasonable amount of variance explained by the model. The overall model test yielded an χ^2^ value of 128 with 21 degrees of freedom, which was statistically significant (*p* < 0.001), indicating that the model significantly predicted the outcome variable (Table 1). In further analysis, we tested for interactions between key variables in our regression model to examine whether the effect of one variable on the outcome was modified by another.

Specifically, we explored interactions between variables such as maternal age and family income, as well as between birth order and maternal health factors (e.g., history of infections during pregnancy). However, none of these interactions were statistically significant, suggesting that the effects of these predictors on child functionality were independent of each other (all *p* ˃ 0.05).

Based on the collinearity statistics, there is no severe multicollinearity in the model, and the predictors are sufficiently independent of each other to produce stable regression coefficients (Appendix A). The predictive measures of the model indicated an overall accuracy of 72.6%, with a specificity of 85.7%, a sensitivity of 51.0%, and an area under the curve (AUC) of 0.79. These results suggest that the model had a good discriminatory ability between high and low/moderate functionality in children with ASD (Appendix A).

## 4. Discussion

The occurrence of ASD has been linked to a range of prenatal, perinatal, and neonatal factors. An increasing number of studies have identified several perinatal and postnatal factors that may be causally associated with ASD, such as fetal distress [13], multiple pregnancy [13], preterm delivery [12,28], meager birth weight [12,13,14,21], and low Apgar score [13].

The present study examines the various factors impacting the functionality of children with ASD through the use of statistical methods and extensive data. The study offers comprehensive descriptive statistics for both continuous and categorical variables. The primary focus of this research is on demographic and physical health variables such as the age of parents at the time of conception, gestational age, and birth weight of the child. For instance, the study highlights the broad range of pre-pregnancy weights among mothers, drawing attention to the diverse physical conditions that existed before pregnancy. With regard to categorical variables, the sociodemographic data indicate that the majority of children come from dual-parent households, and the distribution across different educational levels and incomes suggests a diverse sample.

This research examining high-functioning children with those possessing low or moderate functionality in ASD disclosed multiple noteworthy findings. The key predictors identified through the binomial logistic regression model were maternal age, age at diagnosis, family income, birth order, and mother’s family history with autism, developmental disorders, epileptic seizures, depression, or anxiety disorder.

A relationship was revealed between increased parental age and reduced cognitive functionality in the offspring. For each additional year in the mother’s age, the probability of the child displaying low or moderate cognitive abilities rises by 6%. While this effect is statistically significant, the increase is relatively small. This finding implies that older maternal age and age at conception may pose potential risks. Advanced maternal age, particularly in mothers over 40, is associated with a higher likelihood of severe autism, and this effect is likely driven by age-related genetic mutations, such as de novo mutations, which are more common as women age. Additionally, older mothers are more prone to complications during pregnancy, such as gestational diabetes, hypertension, and preeclampsia, which can adversely affect fetal development. Early intervention programs could be tailored to address specific challenges faced by older mothers.

Prior research [35] has shown that older parental age at the time of birth is correlated with both the severity and incidence of ASD in their progeny. Still, a substantial disparity was discovered in the age of diagnosis, where children with greater functionality were diagnosed at a later age than those with lower functional capacity. For every additional year’s postponement in the diagnosis of ASD, the likelihood of the child having low or moderate functionality diminishes by 35%. This suggests that an earlier diagnosis is associated with higher functionality in children with ASD and may be related to more pronounced ASD symptoms, as the functioning of children may be enhanced with earlier interventions [36,37]. The timing of ASD diagnosis is crucial, with earlier diagnoses (≤3 years) often associated with more severe cases. This trend likely occurs because children with more pronounced symptoms are identified and diagnosed sooner due to the visible nature of their developmental challenges. In contrast, children with milder symptoms may not be diagnosed until later, when subtle differences in behavior become more noticeable. This pattern reflects the fact that severe autism tends to present itself in a way that prompts earlier clinical attention, while milder forms might be overlooked until school age or later when social and communication issues become clearer.

The binomial logistic regression analysis identified several variables that impact functionality, such as vaginal bleeding, infections, socioeconomic factors, and birth order. Vaginal bleeding during pregnancy, especially in the first and third trimester, emerged as a strong predictor of reduced functionality. Mothers who experienced vaginal bleeding in the first trimester were found to have 543% higher odds of having children with low/moderate functionality. Similarly, mothers who experienced vaginal bleeding in the third trimester were found to have 1202% higher odds of having children with low/moderate functionality. Pregnancy complications, whether they occur early or late in gestation, have been consistently linked to reduced functional outcomes. Notably, other surveys have also reported significantly higher rates of bleeding during pregnancy among mothers of autistic children [38], particularly during the mid-trimester [39].

The relationship between prenatal infections caused by viruses or bacteria and cognitive outcomes in offspring is complex and may exhibit paradoxical results. Despite this complexity, our research has shown a substantial decrease in the probability of reduced functionality in children born to mothers who encountered these infections during their pregnancy. Specifically, these children showed a 51% decrease in the probability of having low/moderate functionality. The somewhat counterintuitive results observed in this study may be related to the influence of immune responses or other biological processes on cognitive development. A similar study found no connection between maternal inflammation and ASD [40]. However, evidence suggests that infections reported during pregnancy are associated with neurodevelopment disorders, particularly ASD [41]. Perinatal complications, particularly maternal infections, and vaginal bleeding, are critical risk factors for severe autism, emphasizing the importance of maternal health in prenatal care. This finding indicates that prenatal complications may have long-term effects on neurodevelopment. For obstetricians and paediatricians, this result highlights the need for careful monitoring of pregnancies with complications like vaginal bleeding. Early and proactive management of these complications may help reduce the risk of developmental issues in children. Pregnancies with such complications may warrant additional follow-up and early screening for developmental delays. Our study unveils that the relationship between prenatal infections and cognitive outcomes is multifaceted and intricate, encompassing a range of potential biological factors. While further research is required to fully understand these complex interactions, our results highlight the significance of considering the possible long-term consequences of prenatal infections on offspring cognitive development.

The relationship between family income and functional outcomes in children is of significant interest. A lower family income has been associated with diminished functional outcomes, which may suggest inequities in healthcare access or the availability of early intervention services. According to our data, families with annual incomes of less than EUR 10,000 had a 158% higher likelihood of having children with low/moderate functionality compared to those with incomes ranging from EUR 10,001 to EUR 20,000. This finding highlights the connection between socioeconomic status and adverse functionality outcomes, as families with higher incomes may have greater access to specialized mental health services and more extensive social support. Moreover, studies have confirmed a positive relationship between socioeconomic disparities and the prevalence of children with ASD. [42]. Socioeconomic factors further compound this, with children from lower-income families being significantly more prone to severe autism, possibly due to reduced access to healthcare and early interventions. From a clinical perspective, this finding also underscores the importance of addressing socioeconomic disparities in healthcare access. Children from lower-income families may not receive timely diagnosis or intervention, further exacerbating their challenges.

Yet, the relationship between socioeconomic status, particularly low income, and the presence of severe diagnoses in children, such as ASD, presents a complex and potentially bidirectional dynamic. Parents, often mothers, may find themselves needing to reduce working hours or cease employment entirely to care for a child with severe developmental conditions. This caregiving burden can lead to a significant reduction in family income, thereby exacerbating the financial strain and potentially worsening the family’s overall socioeconomic position. Future research should aim to employ longitudinal designs that can more effectively discern the temporal sequence of these factors, offering clearer insights into causality and better informing interventions aimed at supporting affected families. Policymakers should consider developing programs that provide affordable or subsidized services for low-income families, ensuring that all children with ASD have access to the care they need.

Birth order adds complexity, as second-born children show higher odds of severe autism, which may point to both biological and environmental factors. This also suggests that second-born children may be more vulnerable to developmental challenges, potentially due to differences in parental attention or environmental factors in larger families. Clinically, this finding indicates that healthcare providers should consider family dynamics when assessing developmental risks. Parents of multiple children, particularly those with fewer resources, may require additional support to ensure that all children, including second-born and later-born children, receive adequate developmental care. The status of being a second-born child correlated with an increased probability of exhibiting lower functionality when compared to first-borns, suggesting potential dilution effects of resources or parental attention. Specifically, second-born children demonstrated a 171% heightened likelihood of manifesting low/moderate functionality relative to their first-born counterparts. These data underscore the significant impact of birth order, where subsequent children are more prone to experiencing lower levels of functionality. A comparable study also revealed that the size of the family and the increasing birth order in children with ASD were correlated with heightened functional and cognitive limitations [43].

Interestingly, neonatal indicators, such as crying immediately after birth, show a protective effect, with crying linked to better functionality outcomes. Clinically, this finding underscores the importance of neonatal assessments. Babies who do not cry immediately after birth may require closer monitoring for developmental concerns, and parents should be informed about the potential importance of these early indicators. Neonates who manifested immediate crying following birth demonstrated a 61% reduction in the likelihood of possessing low/moderate functionality in the present study. This observation indicates that prompt vocalization post-delivery, a marker of neonatal vigor and health, correlates with more favorable functional outcomes in later life. Research findings reinforce the potential of crying as a biomarker of an infant’s physical and emotional health status [44]. Yet, studies employing observation of vocalization behaviors post-delivery have been very limited, reducing the external generalization of the findings.

Furthermore, it is noteworthy that individuals with a family history of autism or related developmental disorders exhibit a 42% lower likelihood of experiencing low to moderate functionality, implying that inherited or familial elements may provide a protective influence. A family history of autism or related conditions markedly decreases the likelihood of milder forms of ASD, highlighting a genetic predisposition toward more severe outcomes. However, the univariate analysis of our data presents a seemingly contradictory result: children with a family history of autism or related disorders are more likely to have mild autism. This would suggest that children with a family history are more likely to have better functionality outcomes (mild autism), which appears to contradict the findings of the logistic regression. The discrepancy between the logistic regression and univariate analysis likely stems from the multivariate nature of the regression model. Logistic regression accounts for the influence of multiple factors simultaneously, such as socioeconomic status, maternal age, pregnancy complications, and birth order, which might not be isolated in the univariate analysis. The family history variable may be interacting with other factors, such as early diagnosis or access to interventions, which may moderate the overall effect on functionality.

In contrast, the univariate analysis only compares the raw frequencies of severe vs. mild autism outcomes between those with and without a family history, without controlling for these additional variables. Therefore, the univariate analysis could be masking the underlying complexities that the regression model reveals, particularly the cumulative risk posed by family history when combined with other contributing factors. It is also possible that the maternal family history of conditions like epilepsy, depression, or anxiety may reflect genetic predispositions that affect neurodevelopment in a way that could exacerbate the severity of autism symptoms. Clinicians should take maternal family history into account during prenatal and postnatal care, offering more intensive monitoring and intervention for children born to mothers with a history of developmental or psychiatric disorders. Our findings could be important also for genetic counselling and understanding potential risks in families with a history of developmental or psychiatric disorders. Early intervention programs can be particularly beneficial for these families, potentially mitigating the risk of lower functionality in children with ASD.

Nevertheless, this protective effect is not uniformly supported across the literature. A contrasting study [45] suggests that parental psychiatric disorders, particularly those affecting mothers, may increase the risk of ASD in their children. This risk can be attributed to epigenetic mechanisms that may occur during fetal development, indicating that maternal psychiatric conditions could have a deleterious impact on the neurodevelopment of the offspring [46]. These seemingly contradictory findings underline the complexity of genetic and environmental interactions in the development and manifestation of ASD. While some familial factors may confer protection and contribute to better functionality, others, particularly related to maternal mental health, may increase the risk of developing ASD through mechanisms that alter gene expression during critical periods of prenatal development. This dual perspective calls for a nuanced understanding of the multifactorial nature of ASD, recognizing that both protective and risk factors can coexist within the familial and genetic context [47,48,49].

In interpreting the findings of our study within the broader epidemiological context, it is essential to consider the recent data reported by Kouznetsov et al. (2023), which highlights a prevalence rate of 0.94% for ASD among the 2–17-year-old population in Greece. This prevalence is distinctly higher in males (1.44%) compared to females (0.41%), with a statistically significant odds ratio of 3.55 (CI [3.42, 3.69], *p*-value < 0.001). Our study’s regional analysis indicates a slightly higher prevalence rate in urban areas compared to the national average, which might be attributed to factors such as better access to diagnostic services or greater awareness of ASD symptoms among healthcare providers in urban settings.

Moreover, while the national data reflect substantial gender disparity in ASD diagnoses, our study found a somewhat narrower gap. This discrepancy could be influenced by recent shifts towards more nuanced diagnostic criteria or increased awareness and reporting of ASD in females, who historically have been underdiagnosed due to less overt symptomatology or diagnostic overshadowing.

The total number of individuals diagnosed with ASD, as reported by Kouznetsov et al. [29], stands at 15,706 for the age group of 2–17 years, which underscores the significant public health implications of this condition in Greece. Our findings support these figures but suggest regional variations that may necessitate localized public health strategies and resource allocation to address these disparities effectively [30].

Upon evaluating the study’s strengths, it can be asserted the comprehensive sample size of 517 mothers of ASD-diagnosed children aged between 4 and 7 years provides substantial information on maternal history, pregnancy, childbirth, and neonatal data. Moreover, the participating centers’ services are accessible to children and families from both public and private schools, ensuring a diverse sample without institutional barriers. Additionally, the study’s design considers socioeconomic status, effectively addressing potential geographical disparities and variations. The utilization of a stratified sampling approach, which ensured representation from various regions and demographic groups, further enhanced the generalizability of the findings. The inclusion of multiple peri- and postnatal variables, such as exposure to environmental factors, medical complications during pregnancy, and neonatal health indicators, facilitates a more comprehensive analysis of risk factors associated with ASD functionality. The regression model demonstrated adequate predictive power, with reasonable sensitivity and specificity. The area under the curve (AUC) was robust, indicating the model’s good discriminatory ability between high and low/moderate functionality levels.

This study offers valuable insights into ASD functionality factors, yet several methodological limitations warrant attention. The sample despite being substantial, included only mothers, potentially introducing bias, due to the absence of paternal perspectives. The self-reported data lack independent verification through medical, genetic, or psychiatric assessments, potentially affecting consistency with existing literature. Future research should use larger, more diverse samples and include clinical assessments. While some findings align with current literature, discrepancies may stem from the specific sample and methodology, including reliance on self-reported data, the absence of clinical assessments, and the limited number of input variables, which may limit generalizability and contribute to variations compared to studies incorporating medical and genetic evaluations.

Another limitation of this research is the potential incomplete capture of ASD cases, especially those who did not seek care at the research facilities. This could exclude individuals with milder symptoms or higher functionality. Another constraint relates to data quality, which was influenced by the accessibility of family records and the extent of awareness and engagement from participating centers in the research.

Lastly, it should be noted that the study did not independently verify the ASD diagnoses, but rather relied on records provided by the participating centers. Due to the unavailability of population characteristics, direct tests for sampling bias through comparison of sample distributions to population data could not be conducted. To address potential imbalances in sample proportions, key demographic variables (e.g., gender, and income level) were included as covariates in the regression model. Furthermore, a Chi-square test for proportion differences was performed across all independent variables to assess whether the proportions of different groups (e.g., males vs. females, high income vs. low income) in the sample significantly deviated from the expected distribution.

The results of the study indicate the importance of developing targeted services focused on early intervention, rehabilitation, social care, and integration. It is recommended that policymakers prioritize early diagnosis and accessible early identification and intervention services. Our findings underscore the complex interplay of genetic, environmental, and socioeconomic factors in determining the outcomes of ASD.

These results have significant implications for developing effective early screening and intervention strategies. They emphasize the necessity of considering a wide range of influences to improve the functional outcomes of children with ASD. The detailed statistical analysis conducted in this study offers important insights into the predictors of ASD functionality, providing avenues for targeted interventions and policy adjustments.

Systematic screening of toddlers and increasing awareness among parents, guardians, and healthcare professionals are crucial. Standardizing case-recording procedures and creating an ASD register at the national and regional levels would facilitate systematic monitoring and evaluation of resources and services.

## 5. Conclusions

This research constitutes a comprehensive investigation of ASD in Greece, intending to evaluate the prevalence of ASD at both the national and regional levels, utilizing the functional capacity of children as a basis for assessment. The present study establishes one of the limited contemporary studies that furnishes data essential for estimating the disorder’s burden and informing the formulation and implementation of regional and national service development plans. The study employs statistical analyses of an extensive dataset, which includes various demographic and physiological variables, to explore the factors that influence the functionality of children with ASD. The dataset encompasses predictors such as maternal age, the timing of ASD diagnosis, family income, and birth order.

Several factors, including advanced maternal age, delayed ASD diagnosis, lower family income, and higher birth order, were linked to reduced functionality in children with ASD. The study highlighted significant associations between older maternal age, delayed diagnosis, and poorer functional outcomes, highlighting the need for early intervention. Additionally, children from lower-income families and those with higher birth order in larger families exhibited significantly lower functionality, possibly due to diluted parental resources and socioeconomic disparities. The research also identified complex interactions between prenatal health concerns and child development, with prenatal issues like vaginal bleeding and infections potentially having paradoxical effects through immune responses. Interestingly, a family history of neurological or psychiatric conditions appeared to protect functionality in children with ASD. The regression model employed demonstrated strong predictive power, enhancing the accuracy of distinguishing functionality levels in children with ASD.

The present study sheds light on the significant role that prenatal and early life factors play in determining the functionality of individuals with ASD, thus providing evidence-based information that is essential for estimating the global burden of the disorder and guiding the development of national and regional services. The results of this research underscore the importance of early diagnosis and intervention, systematic screening, and addressing disparities in service provision to improve outcomes for individuals with ASD.

## Figures and Tables

**Figure 1 healthcare-12-02057-f001:**
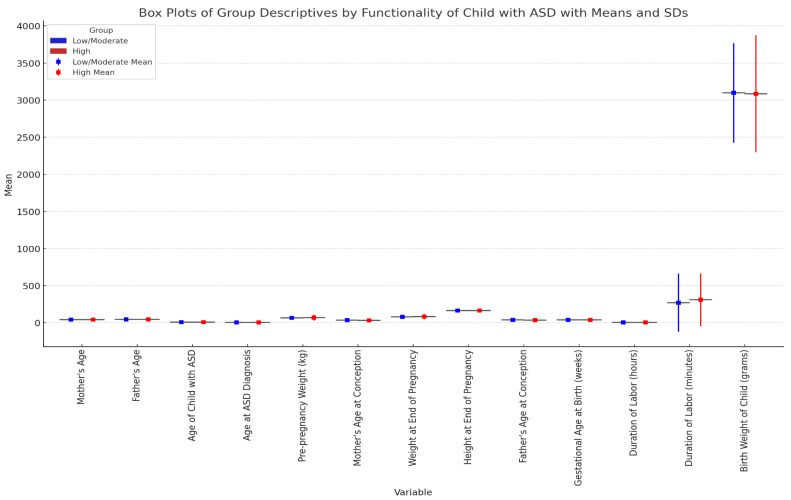
Box plots of group descriptives by functionality of child with autism spectrum disorder (ASD) showing means and standard deviations.

**Figure 2 healthcare-12-02057-f002:**
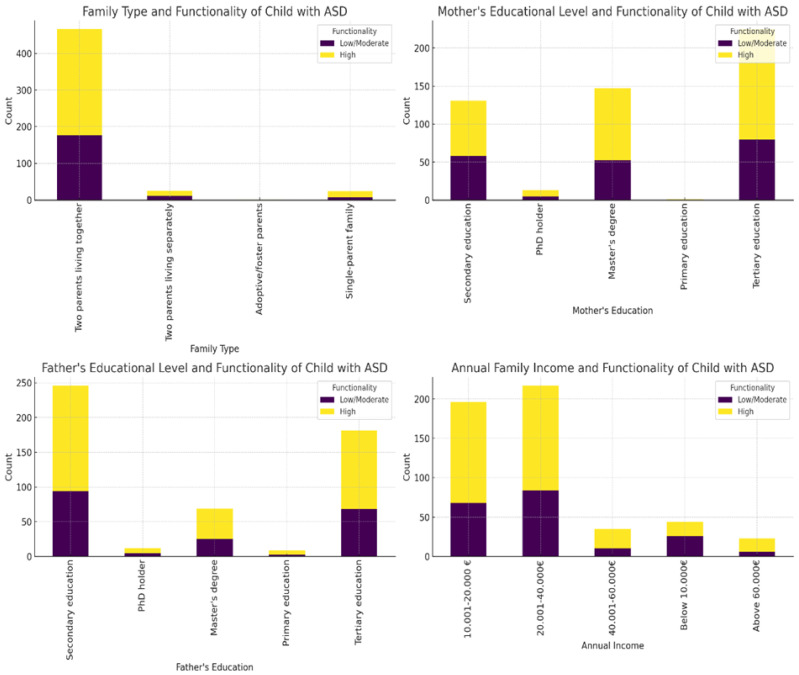
The visualization of the Chi-square test results for the sociodemographic variables. Note: The bar charts depict the association between sociodemographic factors and functionality in children with ASD. Purple represents low/moderate functionality, while yellow represents high functionality. Two-parent households have the majority of children, especially those with high functionality. Higher functionality is more common among children of mothers and fathers with tertiary education, while lower functionality is more prevalent among those with secondary education. Higher family income, particularly in the EUR 20,001–40,000 range, is associated with higher functionality in children, while lower-income families tend to have children with lower functionality.

**Figure 3 healthcare-12-02057-f003:**
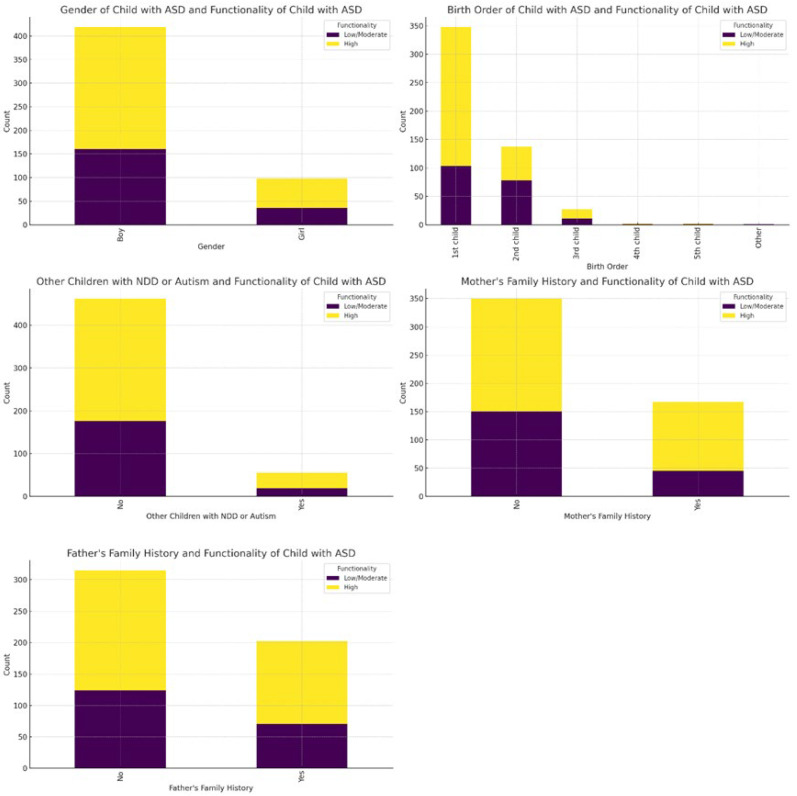
The visualization of the Chi-square test results for the ASD-related variables. Note: The bar charts illustrate the relationship between demographic and family history factors and the functionality of children with ASD. Boys, predominantly showing low/moderate functionality, are more prevalent in both categories, while girls, though fewer, also mainly exhibit low/moderate functionality. First-born children are primarily high-functioning, while later-born children more often have low/moderate functionality. Families without other children diagnosed with neurodevelopmental disorder (NDD) or autism have a higher proportion of high-functioning children. In contrast, those with diagnosed children tend to have more low/moderate functionality children. Maternal and paternal family histories of NDD, autism, or psychiatric disorders are associated with lower functionality in children, whereas their absence is linked to higher functionality.

**Figure 4 healthcare-12-02057-f004:**
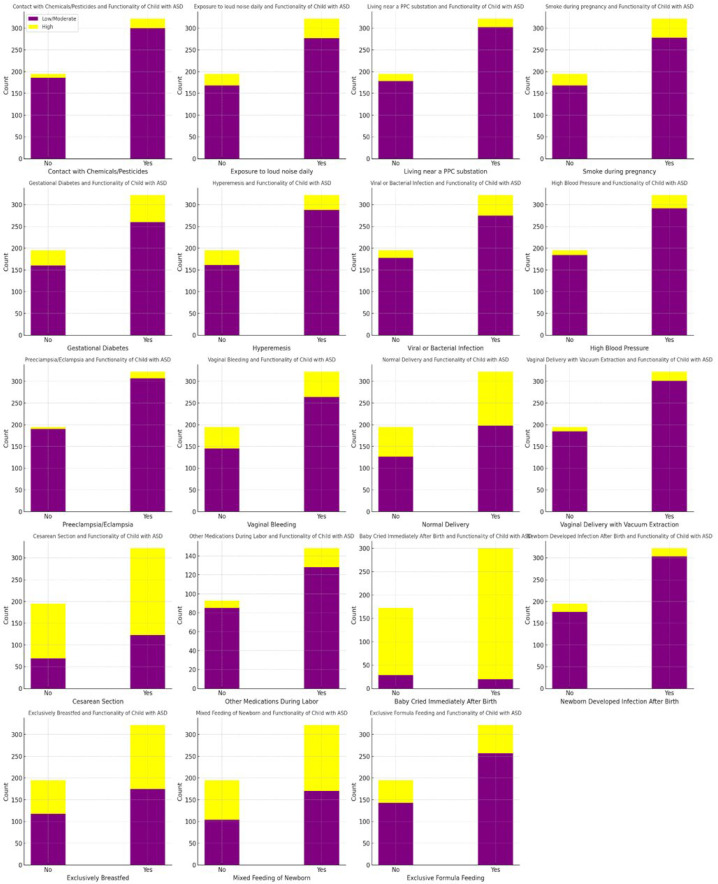
The visualization of the Chi-square test results for the peri- and postnatal variables. Note: The figure presents Chi-square test results for peri- and postnatal factors affecting children with ASD. Each bar chart contrasts the proportion of children with high functionality (yellow) and low/moderate functionality (purple) based on the presence or absence of specific factors. Chemical/pesticide contact, loud noise exposure, and living near PPC substations are associated with low/moderate functionality in children. Maternal smoking during pregnancy, gestational diabetes, hypertension, preeclampsia/eclampsia, vaginal bleeding, cesarean section, and vacuum extraction delivery correlate with increased rates of low/moderate functionality. Immediate crying after birth and exclusive breastfeeding are linked to higher functionality compared to formula or mixed feeding.

**Table 1 healthcare-12-02057-t001:** Binominal logistic regression.

					95% Confidence Interval	
Predictor	Estimate	SE	*p*	Odds Ratio	Lower	Upper
Intercept	−0.96607	0.9678	0.318	0.381	0.0571	2.537
Mother’s Age	0.05840	0.0295	0.048	1.060	1.0005	1.123
Age at ASD Diagnosis	−0.43315	0.0842	<0.001	0.648	0.5498	0.765
Mother’s Age at Conception	−0.00537	0.0312	0.863	0.995	0.9356	1.057
Annual family income:						
20.001–40.000 EUR–10.001–20.000 EUR	0.17062	0.2410	0.479	1.186	0.7395	1.902
40.001–60.000 EUR–10.001–20.000 EUR	0.02553	0.4608	0.956	1.026	0.4158	2.531
Below 10.000 EUR–10.001–20.000 EUR	0.94688	0.3867	0.014	2.578	1.2080	5.500
Above 60.000 EUR–10.001–20.000 EUR	−0.28284	0.5782	0.625	0.754	0.2426	2.341
Birth Order of Child with ASD:						
Second child–First child	0.99497	0.2357	<0.001	2.705	1.7039	4.293
Third child–First child	0.39063	0.4539	0.389	1.478	0.6071	3.598
Fourth child–First child	1.3738	1.5224	0.367	3.95	0.1999	78.079
Fifth child–First child	0.2166	1.503	0.885	1.242	0.0653	23.627
Other–First child	14.39851	535.4115	0.979	1.79 × 10^6^	0	Inf
Mother’s Family History with Autism, Developmental Disorders, Epileptic Seizures, Depression, or Anxiety Disorder:						
Yes–No	−0.55327	0.2325	0.017	0.575	0.3646	0.907
Hyperemesis (severe vomiting):						
Yes–No	0.60889	0.3129	0.052	1.838	0.9956	3.395
Viral or Bacterial Infection:						
Yes–No	−0.71088	0.3493	0.042	0.491	0.2477	0.974
Vaginal Bleeding:						
Yes–No	−1.30802	0.7484	0.081	0.270	0.0624	1.172
Vaginal Bleeding during the pregnancy of the child with ASD (trimester):						
First trimester–No	1.86121	0.7276	0.011	6.432	1.5451	26.771
Second trimester–No	1.46171	0.9177	0.111	4.313	0.7140	26.058
Third trimester–No	2.56609	0.9197	0.005	13.015	2.1460	78.931
Baby Cried Immediately After Birth:						
Don’t remember–No	−0.38883	0.4873	0.425	0.678	0.2608	1.762
Yes–No	−0.94559	0.3738	0.011	0.388	0.1867	0.808

Note: Estimates represent the log odds of “Functionality of Child with ASD = Low/Moderate” vs. “Functionality of Child with ASD = High”. Model fit measures: AIC = 598; Cox and Snell’s R^2^ = 0.220; Nagelkerkes’s R^2^ = 0.300; overall model test χ^2^ = 128; df = 21; *p* ≤ 0.001.

## Data Availability

Due to the sensitive nature of the data, it is not publicly available.

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
