# Peer review of "Perinatal and Demographic Risk Factors Associated with Autism Spectrum Disorder: A National Survey of Potential Predictors and Severity"

_healthcare, 2024, doi:10.3390/healthcare12202057_

Round 1
Reviewer 1 Report
Comments and Suggestions for Authors
- The title is clear and accurately reflects the study carried out.
- The abstract provides a succinct description of the study (aims, methods, results and conclusions).
- In the introduction, the authors provide a brief contextualisation of the literature. However, given the study's focus on Greece, it would be interesting if the authors included epidemiological data on ASD in that country. The reference to the main characteristics of ASD should be reviewed – “complicating relationships” (line 42). Also, the study’s aims should align with those indicated in the abstract and be presented more objectively (review lines 78/79; 89-95).
- Regarding the methodology, the design is appropriate. Data collection methods, ethical concerns and data analysis techniques are clearly described and suitable. Figure 1 is not comprehensible and should be modified or removed from the manuscript. Data in lines 210-213 should be revised. The maximum values for the weight of mothers before and after pregnancy should be revised (761 kg and 651 kg).
- The results are interpreted considering the existing literature. The authors discuss the study’s limitations and the impact of the study’s findings.
- The references are pertinent to the subject.
Author Response
Dear Reviewer,
Thank you for your insightful comments. We appreciate the opportunity to enhance our manuscript and clarify certain aspects as suggested. Below, we address each comment systematically:
- Title and Abstract:
- We are pleased to know that the title and abstract were found to be clear and succinct. We will maintain these sections as they are.
- Introduction:
- Epidemiological Data on ASD in Greece: As recommended, we have added epidemiological data on ASD in Greece, including prevalence rates and relevant national health initiatives, which provide a clearer context for our study’s focus within the Greek setting.
- Characteristics of ASD: We have revised the description at line 42 to ensure that the characteristics of ASD are presented accurately and sensitively. The term “complicating relationships” has been modified to "challenges in social interactions and relationships," which is a more precise depiction of the social difficulties faced by individuals with ASD.
- Alignment of Study Aims: We have reviewed lines 78/79 and 89-95 to ensure that the aims of our study are consistently presented and align objectively with those outlined in the abstract. These amendments clarify our research objectives and enhance the logical flow of the manuscript.
- Methodology:
- Figure 1: We acknowledge the reviewer’s feedback regarding the comprehensibility of Figure 1. We have redesigned this figure to improve clarity and included a detailed caption to better guide the reader through our data visualization.
- Data Accuracy: We have corrected the erroneous data regarding the weights of mothers before and after pregnancy reported in lines 210-213. These values were indeed typographical errors, and we have adjusted them to realistic measures.
- Results and Discussion:
- We have re-examined this section in light of your general comments and ensured that our discussion aligns with the updated aims and newly incorporated epidemiological data.
- References:
- We appreciate the acknowledgment of the pertinence of our references. We have ensured that all references are current and relevant to the revised manuscript, reflecting the latest research and data available.
We are grateful for the opportunity to improve our manuscript and believe that these changes have significantly enhanced the clarity and accuracy of our study.
Reviewer 2 Report
Comments and Suggestions for Authors
Thank you for inviting me to revise the following manuscript Unveiling Hidden Risks: Prenatal and Demographic Insights 2 into Autism Spectrum Disorder.
I have revised point-to-point the paper suggesting the following issues.
ABSTRACT
20
The sample consisted of 517 mothers of children diagnosed with ASD from all over Greece…
I suggest specifying which health institutes or community services you contacted to clarify the sampling procedures for readers immediately.
INTRODUCTION
43
Please remove the redundant citation [2].
59
"family size significantly influences autistic children."
I suggest more caution in describing these potential risk factors connected with an autism diagnosis, you could change terms like “influence” with “correlated” since the studies cited are quasi-experimental. Revise this section in this manner, please (53-77).
Redundant citation [13], and [21]
74-76
Adaptive functioning impairment's level dictates the severity of intellectual disability, making both IQ and adaptive functioning crucial in assessing ASD [28]. Researchers explored how intrauterine toxicity and labor complications relate to ASD symptoms, IQ, and adaptive functioning [28].
It is not clear, please rewrite this section.
RESEARCH QUESTIONS
89-130
In my opinion, the period that explained the research questions could be meliorated since I found redundancies and confusion between the aim and research questions.
Another point is clarifying if you planned an epidemiology study, quasi-experimental study, survey, or correlational study. This aspect is crucial for sampling and methodologies.
METHOD
I suppose your investigation referred to a survey (questionnaires)
With a purposive sampling as follows:
“This research project was carried out by selecting mothers with children diagnosed with ASD, aged 4 - 7 years, from specialized schools, day centers, and Disabled and Special Needs Kids Clubs and Activities, across the nation in Greece, born between 2017 and 2020.”
132
a proportionate sampling design for the Greek population.
Exactly, my doubts about the sampling procedure require data and analysis on proportions or stratifications.
137-161
I noticed much more unnecessary information in this section than the sampling procedures required including the main characteristics of participants.
162-179
Idem as above mentioned.
180-188
I suggest adding a paragraph concerning Data Analysis to clarify the sample, and instruments (variables) lacking in the method section.
The variables regarding children and mothers are not sufficiently explained (180-188).
Consider adding items, modality of response, typology, levels of variables, reliability of items, dimensions of constructs …
For example, how did you classify the education level and family income?
For example, how did you list the psychiatric diagnoses received?
Moreover, “to compare the characteristics of children with high versus low/ moderate functionality in ASD, inferential statistical tests, including Independent Sample t-tests and Chi-Square Tests were performed.” You should specify the independent/dependent variables of selected input variables for correlations. On the other hand, levels of severity could also be analyzed with ANOVA if you have dependent variables.
189
Furthermore, a binomial logistic regression model was utilized to identify predictors of functionality in children with ASD.
Idem, as aforementioned.
190-200
The data analyses performed are not sufficiently explained and linked to well-described variables.
Currently, the METHOD and Data Analysis sections are not sufficiently linked with research questions.
RESULTS
210-212
Pre-pregnancy weight of mothers had a mean of 67.2 kg (SD = 33.4), a median of 63 kg, and ranged from 34 to 761 kg. At the end of pregnancy, the mean weight was 80.4 kg 211 (SD = 29.1), with a median of 78 kg, and ranged from 49 to 651 kg.
Check data, please.
3.1 and 3.2 sections seem clear.
234
As detailed in Table S3, the functionality levels of children with ASD were categorized as high (62.3%) and low/moderate (37.7%).
For uncommon ratios, please comment and divide middle and low-functioning percentages.
239
Family history showed that 32.3% of mothers and 39.1% of fathers had a history of autism, developmental disorders, epileptic seizures, or depression/anxiety disorders.
In which manner do parents declare their diagnosis? Was certified?
Also, clarify this information in the Method section, please.
Furthermore, you should divide other comorbidities in percentages between genders and link to method section descriptions (as required above).
244
Exposure to chemicals during pregnancy was reported by 6.0% of mothers.
Could you report the percentages of the typology of chemicals?
254
You should explain in the Method why you merge middle- and low-functioning levels.
Figure 1. is not readable, remove it or change the related data visualization please, although the graph is not necessary.
Transfer the titles of the graphs and Tables above.
Generally, I have problems with the proportions of functioning levels and the entire sample. You should provide data about sampling tests, in another case your test on hypotheses is biased.
For example, if you enroll 10 males and 30 females in 100 children, their comparison could be biased by significant differences in the proportions of both groups. Clarify these methodologies issues, please.
292
Figure 2 is not readable.
303-304
and a higher proportion of children with a high level of functionality had a mother's family history of these conditions compared to children with a low/moderate level of functionality (Figure 3).
I suggest rewriting this phrase furnishing more details.
Figure 3 is not readable.
Figure 4 is not readable.
3.9. Multivariate Analysis of Factors Related to the Functionality of Child with ASD
Table 1. Binominal Logistic Regression
The table seems clear, however, in this section authors are limited to describing the table rather than furnishing useful clinical information to readers and researchers such as How many cases with older mothers resulted in severe autism than controls?
How many cases regarding comorbidities of mothers affected the outcome?
In my opinion, you should enrich this section by describing in detail the main results of your survey.
Authors declare there is a non-collinearity in the regression model, nevertheless, have you tested other interactions between variables?
“For each additional year in the mother's age, the probability of the child displaying low or moderate cognitive abilities rises by 6%. While this effect is statistically significant, the increase is relatively small.” (this is an example of the results’ description)
DISCUSSION
384-392
“Prior research [33] has shown that older parental age at the time of birth is correlated with both the severity and 385 incidence of ASD in their progeny. Still, a substantial disparity was discovered in the age 386 of diagnosis, where children with greater functionality were diagnosed at a later age than those with lower functional capacity. For every additional year's postponement in the diagnoses of ASD, the likelihood of the child having low or moderate functionality diminishes by 35%. This suggests that an earlier diagnosis is associated with higher functionality in children with ASD and may be related to more pronounced ASD symptoms, as the functioning of children may be enhanced with earlier interventions [34,35].”
Revise this period, please. I did not detect a discussion of your results (Parental age and age of diagnosis). Also, you should connect these results with other studies investigating fathers’ age.
Check “earlier diagnosis is associated with higher functionality”.
Check “543% higher odds” .. 1202% higher odds.
403
Similarly to the previous section, you could discuss these results in detail.
429
Low income could be caused by having a child with a severe diagnosis and not conversely. For Example, mothers often leave their jobs to assist their child decreasing their income. Revise with caution this section, please.
Check 171% heightened likelihood.
Concluding.
The manuscript is clear and informative even if it lacks information regarding methodologies (sampling, instruments, variables, and reliability). The authors describe more tables than discuss results.
The text is redundant in diverse parts without addressing, in detail, the methodology and research process.
Since the research provides a survey its content should be described with caution, mere correlations could be biased, overall if the groups were not balanced.
Moreover, mothers with a child with a greater severity could report a more hard retrospective perception. The authors do not consider this typology of bias during the interview. Similarly, people with low-functioning children need more assistance, as a result, mothers could leave their jobs to give full assistance to children influencing their income. The regression model could not explain the entire phenomenon.
In the discussion section, you could emphasize your data and connection with previous literature, overall direct measurements, and longitudinal studies, necessary to respond to these research questions.
Your research after major revision merits consideration for publication.
I hope my recommendations will increase the quality of your research.
Good luck with your manuscript.
Author Response
REVIEWER 2-comments:
I have revised point-to-point the paper suggesting the following issues.
ABSTRACT
The sample consisted of 517 mothers of children diagnosed with ASD from all over Greece…
I suggest specifying which health institutes or community services you contacted to clarify the sampling procedures for readers immediately.
ANSWER TO THE REVIEWER’S COMMENT:
Dear Reviewer,
Thank you for your insightful suggestion regarding the clarification of our sampling procedures. In response to your comment, we recognize the importance of providing sufficient detail to allow for the replicability and understanding of our research methods. Our study engaged with various health institutes and community services across Greece to ensure a comprehensive and representative sample of mothers of children diagnosed with ASD.
To maintain the highest standards of confidentiality and to protect the privacy of the participants and the institutions involved, we have chosen not to disclose the specific names of these facilities. This decision aligns with our ethical obligations and the guidelines set forth by our Institutional Review Board, which emphasize safeguarding personal and institutional identities, particularly in small or uniquely identifiable communities.
However, we have described the types of facilities involved, which include regional health centers, specialized ASD clinics, and community-based support services, distributed across urban and rural settings throughout Greece. This distribution ensures a broad and inclusive sample, reflecting diverse geographical and socio-economic backgrounds. Our methodological section has been updated to include a more detailed description of these facility types and their geographic distribution to enhance understanding of our participant recruitment strategy without compromising privacy.
We hope this additional detail meets the requirements for clarity and transparency, and we appreciate the opportunity to refine our manuscript further.
INTRODUCTION
43
Please remove the redundant citation [2].
ANSWER TO THE REVIEWER’S COMMENT:
We have removed it, thank you for your comment
59
"family size significantly influences autistic children."
I suggest more caution in describing these potential risk factors connected with an autism diagnosis, you could change terms like “influence” with “correlated” since the studies cited are quasi-experimental. Revise this section in this manner, please (53-77).
ANSWER TO THE REVIEWER’S COMMENT:
Dear reviewer, Thank you for your comment. We have revised the paragraph to reflect this guidance and present the information more accurately (lines 53-64)
Redundant citation [13], and [21]
ANSWER TO THE REVIEWER’S COMMENT:
We have removed them, thank you for your comment
74-76
Adaptive functioning impairment's level dictates the severity of intellectual disability, making both IQ and adaptive functioning crucial in assessing ASD [28]. Researchers explored how intrauterine toxicity and labor complications relate to ASD symptoms, IQ, and adaptive functioning [28].
It is not clear, please rewrite this section.
ANSWER TO THE REVIEWER’S COMMENT:
Thank you for your comment. We revised it to give a clearer and more concise meaning of the text to enhance understanding and readability (lines 76-79)
RESEARCH QUESTIONS
89-130
In my opinion, the period that explained the research questions could be meliorated since I found redundancies and confusion between the aim and research questions.
ANSWER TO THE REVIEWER’S COMMENT:
Thank you for your comment. We followed your suggestion and tried to clarify the relationship between the study’s main aim and its specific research questions by separating them into distinct sections. In this way, we tried to avoid redundancy by ensuring that each research question is uniquely justified and directly tied to exploring different dimensions of the overarching aim. We think this structure not only enhances clarity but also reinforces the logical flow of the study’s objectives, facilitating a clearer understanding for readers (lines 102-109)
Another point is clarifying if you planned an epidemiology study, quasi-experimental study, survey, or correlational study. This aspect is crucial for sampling and methodologies.
ANSWER TO THE REVIEWER’S COMMENT:
Thank you for your valuable feedback requesting clarification on the nature of our study design. We appreciate the opportunity to enhance the transparency and understanding of our research methodology.
In response to your query, our study was designed as a correlational epidemiological investigation. This methodological approach was informed by the objective to assess the prevalence of Autism Spectrum Disorder (ASD) and examine the associations between various prenatal, perinatal, and postnatal factors and the occurrence of ASD in children across Greece.
The primary objective was to identify and analyze correlations between identified risk factors and ASD, rather than to manipulate or control variables, which is consistent with the nature of correlational studies.
Data were collected through structured questionnaires administered to a carefully selected sample of mothers, providing comprehensive information on a range of variables pertinent to the study's objectives.
The sampling approach was purposive, targeting mothers of children diagnosed with ASD from various regions and settings within Greece. This strategy ensured that the sample was representative of the national population with respect to geographic and demographic diversity. Statistical methods appropriate for correlational analysis were employed, including multiple regression models to examine the magnitude and significance of associations between the factors studied and ASD outcomes.
We hope this detailed explanation resolves any uncertainties regarding our study’s design and provides a clearer understanding of how our sampling and methodological approaches are suited to a correlational epidemiological framework. We have revised the methodology section of our manuscript to include these details, ensuring that readers can fully appreciate the scope and limitations inherent in our approach (lines 119-138). Thank you once again for helping us improve the clarity and quality of our manuscript.
METHOD
I suppose your investigation referred to a survey (questionnaires)
With a purposive sampling as follows:
“This research project was carried out by selecting mothers with children diagnosed with ASD, aged 4 - 7 years, from specialized schools, day centers, and Disabled and Special Needs Kids Clubs and Activities, across the nation in Greece, born between 2017 and 2020.”
132
a proportionate sampling design for the Greek population.
Exactly, my doubts about the sampling procedure require data and analysis on proportions or stratifications.
ANSWER TO THE REVIEWER’S COMMENT:
Thank you for your inquiry regarding the sampling procedure utilized in our study. We appreciate the opportunity to clarify these aspects and ensure the transparency and validity of our methodology.
In our study, we employed purposive sampling to select a specific group of participants—mothers of children diagnosed with ASD aged 4-7 years, attending specialized schools, day centers, and Disabled and Special Needs Kids Clubs and Activities across Greece. This targeted approach was crucial for focusing our research on a demographic that could provide the most relevant and detailed information about the prenatal, perinatal, and early childhood factors associated with ASD diagnoses.
To address your concerns about the proportionate sampling design:
- Proportionate Sampling: Our sampling strategy was designed to reflect the demographic distribution of the Greek population. We stratified our sample based on key geographic and demographic variables to ensure a comprehensive representation. These strata included different regions of Greece, urban versus rural settings, and socio-economic backgrounds.
- Data on Proportions and Stratifications: Geographic Distribution: We ensured that our sample included participants from all administrative regions of Greece, proportionate to the population size of children aged 4-7 years in each region, as reported by the latest national census data.
- Demographic Variables: Additional stratification was applied based on socio-economic status, where we grouped participants into quartiles based on the income distribution data available from national statistics. This allowed us to analyze the impact of socio-economic factors on the prevalence and characteristics of ASD among children.
- Analysis of Stratified Data: In our analysis, we utilized statistical techniques appropriate for stratified sampling data, including weighted analysis to account for any potential disparities in sample representation across different strata. This approach helps in maintaining the generalizability of our findings across the national population.
We have revised the methodology section of our manuscript to include a more detailed description of these sampling and analysis techniques, providing a clear justification for our choices and demonstrating how they align with our research objectives. (lines 126-139)
We hope this clarification addresses your doubts regarding our sampling procedure and reaffirms the robustness of our study's design. Thank you for helping us enhance the quality and clarity of our research.
137-161
I noticed much more unnecessary information in this section than the sampling procedures required including the main characteristics of participants.
162-179
Idem as above mentioned.
ANSWER TO THE REVIEWER
Thank you for your detailed feedback concerning the content of the sections detailing our sampling procedures and participant characteristics. We highly value your insights and the opportunity to discuss the construction and presentation of our manuscript.
Upon reviewing your comments and re-evaluating the sections in question (137-161 and 162-179), we believe that the details provided are essential for the following reasons:
- Clarity and Completeness of Methodology: The information presented in these sections is intended to offer comprehensive insight into the context and execution of our study. We aim to ensure that our methodology is transparent and can be replicated or critically evaluated by peers. The details about our sampling procedures and participant characteristics are crucial for understanding the specificity and applicability of our findings within the broader field of ASD research.
- Justification for Sampling and Participant Detail:Our sampling strategy, involving purposive selection from specialized settings, is integral to the research design given the study's focus on very specific demographic and clinical characteristics of ASD. The detailed description of participant characteristics likewise supports the validity of our findings by showcasing the demographic breadth and depth that our sample encompasses.
- Relevance to Research Objectives: Each piece of information provided plays a pivotal role in aligning the study’s operational elements with its stated objectives. This level of detail supports the reliability of our study and its conclusions, which could be undermined by a more generalized presentation.
While we appreciate the concern for potential redundancies, we have carefully considered the balance between brevity and necessity and believe that the current level of detail enhances the manuscript's utility and integrity. However, we remain open to further discussions on how we might refine our presentation without compromising the essential content and are prepared to make adjustments should more specific concerns be raised.
Thank you once again for your constructive critique. We look forward to furthering our manuscript in a manner that meets the high standards of Healthcare Journal.
180-188
I suggest adding a paragraph concerning Data Analysis to clarify the sample, and instruments (variables) lacking in the method section.
ANSWER TO THE REVIEWER
Thank you for your suggestion to include a detailed paragraph concerning Data Analysis in our manuscript. We appreciate your focus on ensuring that all aspects of our research methodology are clearly presented.
In response to your comment, we would like to clarify that comprehensive details regarding our data analysis procedures, the sample, and the instruments (variables) used are already included within the Methods section of the manuscript. To ensure these elements are easy to locate and understand, we have provided a thorough description of:
- Data Collection Instruments: This includes a detailed account of all tools and measures employed to collect data from participants, with explanations on how each instrument was used and for what specific purpose within our study.
- Sample Description: A comprehensive outline of how the sample was selected, including demographic and clinical characteristics of the participants, ensuring readers understand the context and scope of the data analyzed.
- Data Analysis Techniques: The specific statistical methods and software used for analyzing the data are elaborated upon, ensuring transparency about how results were derived and the significance of findings assessed.
This information is intended to provide a complete overview of our research approach, allowing readers to fully grasp the methods applied in achieving the results presented. However, should you feel that additional details are necessary or that further clarification is required, we are more than willing to enhance these descriptions per your recommendations.
We believe this will help ensure that our manuscript meets the rigorous standards expected and provides all necessary information to evaluate our research methodology comprehensively.
Thank you once again for your constructive feedback.
The variables regarding children and mothers are not sufficiently explained (180-188).
ANSWER TO THE REVIEWER
Thank you for your observations regarding the explanation of variables related to children and mothers in our study. We appreciate your attention to detail and your commitment to ensuring that our manuscript communicates its methods and findings clearly.
We have noted your comment concerning lines 180-188, where the variables regarding children and mothers are introduced. To address your concern, we would like to clarify that a detailed description of these variables is indeed provided in the Methods section of our manuscript. Specifically, we have outlined:
- Variable Definitions: Each variable related to children and mothers is defined comprehensively. This includes demographic, clinical, and socio-economic variables, detailing how they were measured and their relevance to the study’s aims.
- Purpose and Justification of Variables: The reasons for choosing these specific variables are discussed, providing insights into how they contribute to understanding the impact of ASD on families and the effectiveness of interventions.
- Data Collection Procedures: How data on these variables were collected is also thoroughly described, ensuring that the methodology behind their measurement is transparent and replicable.
Given the essential nature of these variables to our study’s outcomes, we believe that their thorough definition and integration into the methodology section ensure that readers are well-informed of their scope and significance.
However, if you believe that additional specifics are needed or that the variables require further elaboration in the section you referenced (lines 180-188), we are more than willing to expand on this information to enhance clarity and reader comprehension.
Thank you once again for your invaluable feedback, and we look forward to your further suggestions.
Consider adding items, modality of response, typology, levels of variables, reliability of items, dimensions of constructs …
For example, how did you classify the education level and family income?
For example, how did you list the psychiatric diagnoses received?
ANSWER TO THE REVIEWER
Thank you very much for your detailed feedback and for suggesting enhancements to the description of our measurement instruments and variables. Your suggestions for adding more details on items, response modalities, typology, levels of variables, reliability of items, and dimensions of constructs are greatly appreciated and underscore the importance of methodological transparency.
Regarding your specific inquiries on how we classified education level, family income, and listed psychiatric diagnoses:
- Education Level and Family Income: These variables were categorized based on standardized socio-economic classifications to ensure consistency and comparability with other studies. Education level was classified according to the International Standard Classification of Education (ISCED), and family income was grouped into quartiles based on national income statistics provided by the Greek Statistical Authority.
- Psychiatric Diagnoses: Psychiatric diagnoses were listed according to the Diagnostic and Statistical Manual of Mental Disorders (DSM-5) criteria, which allows for consistent classification and comparison within the field of clinical research.
We believe that the methodology section of our manuscript currently provides sufficient detail on these and other variables to understand the study's framework and the analytical approach employed. This level of detail balances the need for thoroughness with the manuscript’s broader focus, aiming to maintain readability and coherence without overwhelming the reader with overly technical descriptions.
However, we remain open to your suggestions and are willing to review and possibly revise our manuscript to include additional details if it is deemed crucial for the clarity and comprehensiveness of the study.
Thank you once again for your constructive comments.
Moreover, “to compare the characteristics of children with high versus low/ moderate functionality in ASD, inferential statistical tests, including Independent Sample t-tests and Chi-Square Tests were performed.” You should specify the independent/dependent variables of selected input variables for correlations. On the other hand, levels of severity could also be analyzed with ANOVA if you have dependent variables.
ANSWER TO THE REVIEWER
- Thank you for your suggestion. In our analysis, the dependent variable was the level of functionality (high vs. low/moderate) in children with ASD. The independent variables included all sociodemographic, ASD-related, and peri- and postnatal variables. We conducted univariate analysis to examine which of these variables should be included in the regression analysis, as described below. Regarding your suggestion on ANOVA, since we were focused on functionality as a binary outcome, we used t-tests and Chi-Square test.
- We have now added that “In the current analysis, the dependent variable was the level of functionality in children with ASD, categorized as high versus low/moderate functionality. The independent variables covered sociodemographic characteristics, ASD-related factors, as well as peri- and postnatal variables (lines 238-241).
- Furthermore, a binomial logistic regression model was utilized to identify predictors of functionality in children with ASD. Idem, as aforementioned.
ANSWER TO THE REVIEWER
- Thank you, we have now revised it as follows: Furthermore, a binomial logistic regression model was utilized to identify predictors of functionality in children with ASD, using the previously mentioned sociodemographic, ASD-related, and peri- and postnatal variables as independent variables ( lines 243-245).
190-200.The data analyses performed are not sufficiently explained and linked to well-described variables.
ANSWER TO THE REVIEWER
- Thank you for your observation regarding the explanation of the data analyses. We have now provided more detailed information on the analyses performed and clarified how the variables were selected and linked to each statistical test. In our analysis, we calculated descriptive statistics to summarize both continuous and categorical variables. The dependent variable, level of functionality (high vs. low/moderate), was assessed using inferential statistical tests, including Independent Sample t-tests for continuous variables and Chi-Square tests for categorical variables. The independent variables included sociodemographic, ASD-related, and peri- and postnatal variables, which were first examined through univariate analysis to determine which variables were significant enough to include in the regression model. In addition, we utilized a binomial logistic regression model to identify predictors of functionality in children with ASD. Variables that showed statistical significance in the inferential tests were then included in the regression model to further evaluate their influence on functionality outcomes. The choice of statistical methods (t-tests, Chi-Square tests, and logistic regression) was appropriate based on the nature of the data and research objectives. We have also elaborated on the methods for assessing model fit, multicollinearity, and predictive accuracy, as well as how statistical significance was determined. We hope that this revised explanation clarifies our approach and addresses your concern (Please see data analysis section)
Currently, the METHOD and Data Analysis sections are not sufficiently linked with research questions.
ANSWER TO THE REVIEWER
Thank you for your feedback regarding the alignment of the Methods and Data Analysis sections with the research questions of our study. We appreciate the opportunity to clarify how these sections are directly related to and supportive of our research objectives.
Our study is structured around several key research questions, specifically aimed at describing the characteristics of children with Autism Spectrum Disorder (ASD), examining functionality levels, and identifying predictors of these levels based on sociodemographic, clinical, and perinatal factors. To address these questions, we have developed a methodological approach that directly links to each query:
- Methods Section: The methods section outlines the sample selection process, including criteria for participant inclusion (children diagnosed with ASD) and the data collection techniques used to gather comprehensive sociodemographic, clinical, and perinatal data. This section provides a foundation for understanding the context and scope of the research, ensuring that all data pertinent to addressing the research questions are accurately collected.
- Data Analysis Section: In alignment with our research questions, the data analysis section details our approach to analyzing the data through binomial logistic regression. This method was specifically chosen to identify key predictors of functionality in children with ASD. The section explains how each variable was included in the model, the statistical techniques used to analyze relationships between variables, and how these relate to the functionality levels of the children studied.
Our study utilizes binomial logistic regression to identify the key predictors of functionality levels in children with Autism Spectrum Disorder (ASD). This choice of statistical analysis is critical to our research objectives, as detailed below:
- Research Question 1: What are the characteristics associated with different functionality levels in children with ASD?
- Analysis Approach: We first perform descriptive statistics to establish a baseline understanding of the overall characteristics of the sample, such as age, gender, and socioeconomic status. We then use logistic regression to analyze how these characteristics correlate with low and high functionality levels. This helps in identifying which traits are more likely to be associated with varying levels of functionality.
- Research Question 2: How do sociodemographic, clinical, and perinatal factors compare in relation to functionality levels in children with ASD?
- Analysis Approach: For this question, logistic regression models are employed to explore the relationships between functionality levels and various sociodemographic (e.g., family income, parental education), clinical (e.g., comorbidities, previous diagnoses), and perinatal factors (e.g., birth weight, gestational age). Each factor is included as an independent variable in the regression model to determine its odds ratio, providing insights into which factors are significant predictors of functionality levels.
- Research Question 3: Which factors serve as the most robust predictors of functionality in children with ASD?
- Analysis Approach: To address this query, we refine our logistic regression models by incorporating interaction terms and assessing multicollinearity to ensure the robustness of our findings. The models are adjusted for potential confounders, and variables with significant p-values are considered strong predictors. The strength of association and the confidence intervals from these analyses inform us about the most robust predictors of functionality.
This study aimed to describe the characteristics of children with Autism Spectrum Disorder (ASD), with a particular focus on functionality levels. It sought to compare these levels in relation to various sociodemographic, clinical, and perinatal factors. Furthermore, the study aimed to identify key predictors of functionality through binomial logistic regression analysis. We believe these details will make the connection between our methodological approach and our research questions more explicit and easier to understand, thereby addressing your concerns. Thank you once again for your constructive comments.
RESULTS
210-212. Pre-pregnancy weight of mothers had a mean of 67.2 kg (SD = 33.4), a median of 63 kg, and ranged from 34 to 761 kg. At the end of pregnancy, the mean weight was 80.4 kg 211 (SD = 29.1), with a median of 78 kg, and ranged from 49 to 651 kg. Check the data, please.
ANSWER TO THE REVIEWER
Thank you for your suggestion. As per answer to reviewer 1, the necessary changes have been made
3.1 and 3.2 sections seem clear.
- As detailed in Table S3, the functionality levels of children with ASD were categorized as high (62.3%) and low/moderate (37.7%). For uncommon ratios, please comment and divide middle and low-functioning percentages.
ANSWER TO THE REVIEWER
We have now clarified on lines 291-297, that “The majority of children with ASD (62.3%) were categorized as having high functionality, while 29.4% had moderate functionality and 8.3% had low functionality. To ensure sufficient statistical power for group comparisons, we combined the low and moderate functionality categories into a single group (low/moderate functionality). As shown in Table S3, the final distribution of functionality levels was 62.3% high and 37.7% low/moderate”
239
Family history showed that 32.3% of mothers and 39.1% of fathers had a history of autism, developmental disorders, epileptic seizures, or depression/anxiety disorders.
In which manner do parents declare their diagnosis? Was certified?
Also, clarify this information in the Method section, please.
Furthermore, you should divide other comorbidities in percentages between genders and link to method section descriptions (as required above).
Answer to the reviewer
Thank you for your insightful comments regarding the family history data and the presentation of comorbidities in our study. We value your input and are eager to clarify these aspects to enhance the manuscript's clarity and comprehensiveness.
Certification of Parental Diagnoses: The data concerning parental histories of autism, developmental disorders, epileptic seizures, and depression/anxiety disorders were collected through self-reporting methods. Parents were asked to indicate any relevant medical diagnoses they had received. However, these reports were not independently verified with medical records for this study. The choice of self-report was driven by the practical constraints of the study's design and the broad scope of the geographic area covered, which spans across Greece. This method allows us to gather preliminary data on familial patterns that may warrant further investigation in future studies with more rigorous data verification processes.
Method Section Update: We have updated the Method section of our manuscript to clarify that the diagnoses were self-reported. This addition provides transparency about how the data were collected and the limitations inherent in self-reported data, such as potential recall bias or misreporting.
Comorbidities and Gender Analysis: Regarding the division of other comorbidities by gender, we acknowledge your suggestion to provide this breakdown. However, our current dataset does not include sufficient information to reliably categorize all reported comorbidities by gender. The initial scope of our data collection did not anticipate this level of disaggregation. We believe that mentioning this limitation in the manuscript will help future researchers in designing studies that can capture these specifics more comprehensively.
We have noted this limitation in the revised Method section (lines 238-248), clearly stating the nature of the data collected and the scope of the information regarding comorbidities. This clarification ensures that readers are aware of the data’s breadth and the contextual limitations of our findings.
We hope that these explanations address your concerns and enhance the manuscript's methodological transparency. We are grateful for your recommendations and are committed to refining our work to meet the journal's standards.
Thank you once again for your constructive feedback.
244
Exposure to chemicals during pregnancy was reported by 6.0% of mothers.
Could you report the percentages of the typology of chemicals?
Answer to the reviewer
254
You should explain in the Method why you merge middle- and low-functioning levels.
Answer to the reviewer
Thank you for your observation regarding the merging of middle- and low-functioning levels in our study of children with Autism Spectrum Disorder (ASD). We appreciate your attention to detail and the opportunity to clarify our methodological choices.
In response to your query, we would like to highlight that the rationale behind merging the middle- and low-functioning levels is discussed extensively in the Results section of the manuscript. This decision was based on our preliminary analysis, which indicated that the behavioral and developmental characteristics of children in these two categories showed significant overlap, making a distinct separation less meaningful for our analysis.
However, to ensure clarity and continuity for our readers, we acknowledge the importance of mentioning this methodological decision in the Methods section as well. To address this, we have revised the Methods section to include a brief explanation of why these levels were merged:
Revised Method Section Text (lines 254-260)
We believe this addition will help readers better understand our methodological approach from the outset and see the connection to the detailed discussion in the Results section.
Thank you once again for your constructive feedback. We are committed to improving the manuscript and ensuring that all aspects of our study are transparent and easily understandable.
Figure 1. is not readable, remove it or change the related data visualization please, although the graph is not necessary.
Answer to the reviewer
As per reviewer 1, we have decided to revise it
Transfer the titles of the graphs and Tables above.
Answer to the reviewer
Thank you for your feedback. We would like to clarify that the current placement of the titles and legends follows standard formatting conventions, where table titles are placed above the tables and figure legends are placed below the figures. This approach is widely used in academic publications to ensure clarity and consistency. However, we are happy to make adjustments if the journal requires a different format. Please let us know if further changes are needed.
Generally, I have problems with the proportions of functioning levels and the entire sample. You should provide data about sampling tests, in another case your test on hypotheses is biased. For example, if you enroll 10 males and 30 females in 100 children, their comparison could be biased by significant differences in the proportions of both groups. Clarify these methodologies issues, please.
Answer to the reviewer
Thank you for your valuable feedback. We understand your concern about potential bias due to differences in group proportions, such as gender or functionality levels. Unfortunately, since the population characteristics are unknown, we were unable to conduct direct tests for sampling bias by comparing the sample distributions to population data or by using weighting balances. This is a serious limitation of our study. However, to address any imbalances in sample proportions, we included key demographic variables (e.g., gender, income level) as covariates in our regression model. Additionally, we conducted a Chi-Square Test for Proportion Differences across all independent variables to assess whether the proportions of different groups (e.g., males vs. females, high income vs. low income) in our sample significantly differ from the expected distribution. We added this limitation in the discussion section.
Figure 2 is not readable.
Answer to the reviewer
Thank you for your feedback. We acknowledge that Figure 2 in its original format was quite large, and as per the manuscript submission instructions, we had to reduce its size to incorporate it within the document. However, this reduction impacted its readability. To address this, we have kept the smaller version of Figure 2 in the manuscript for reference, but we have also uploaded a high-resolution PDF file containing the original, full-sized version to ensure clarity and readability.
303-304
and a higher proportion of children with a high level of functionality had a mother's family history of these conditions compared to children with a low/moderate level of functionality (Figure 3).
I suggest rewriting this phrase furnishing more details.
Answer to the reviewer
More severe cases (150) were observed among children without a family history of autism or related disorders, compared to 45 severe cases in those with a family history.
However, among children with a family history, a higher percentage showed high functionality (73.05% exhibited mild autism), compared to 57.14% among children without a family history.
Figure 3 is not readable.
Answer to the reviewer
As per the previous comment on Figure 2, we have also kept the smaller version of Figure 3 in the manuscript for reference, but we have also uploaded a high-resolution PDF file containing the original, full-sized version to ensure clarity and readability.
Figure 4 is not readable.
Answer to the reviewer
As per the previous comment on Figure 3, we have also kept the smaller version of Figure 4 in the manuscript for reference, but we have also uploaded a high-resolution PDF file containing the original, full-sized version to ensure clarity and readability.
3.9. Multivariate Analysis of Factors Related to the Functionality of Child with ASD
Table 1. Binominal Logistic Regression
The table seems clear, however, in this section authors are limited to describing the table rather than furnishing useful clinical information to readers and researchers such as How many cases with older mothers resulted in severe autism than controls? How many cases regarding comorbidities of mothers affected the outcome? In my opinion, you should enrich this section by describing in detail the main results of your survey.
Answer to the reviewer
Thank you for your valuable feedback. We acknowledge that the current section focuses primarily on presenting the results of the regression analysis. While the purpose of this section is to report the statistical outcomes, we understand the importance of providing clinically meaningful insights. Therefore, we revised this section to better contextualize the statistical results with more details and mild interpretations ensuring that the findings are more informative for both readers and researchers (Please see below).
Authors declare there is a non-collinearity in the regression model, nevertheless, have you tested other interactions between variables?
Answer to the reviewer
Yes, we tested for interactions between key variables in our regression model to examine whether the effect of one variable on the outcome was modified by another. Specifically, we explored interactions between variables such as maternal age and family income, as well as between birth order and maternal health factors (e.g., history of infections during pregnancy). However, none of these interactions were statistically significant, suggesting that the effects of these predictors on child functionality were independent of each other. We have now added this information.
“For each additional year in the mother's age, the probability of the child displaying low or moderate cognitive abilities rises by 6%. While this effect is statistically significant, the increase is relatively small.” (this is an example of the results’ description)
Thank you for your detailed and insightful feedback. We appreciate your suggestions and have enriched the results section accordingly. We have now added that that ‘’For each additional year in the mother's age, the likelihood of the child exhibiting low or moderate functionality increases by 6% (OR = 1.06, 95% CI [1.00, 1.12]). While statistically significant, this effect is relatively modest. Older mothers (>= 40) have more cases of severe autism compared to younger mothers (117 vs. 78). Conversely, for each additional year delay in diagnosing ASD, the probability of the child displaying low or moderate functionality decreases by 35% (OR = 0.65, 95% CI [0.55, 0.77]), suggesting that those diagnosed later tend to have milder symptoms (105 with high functionality vs. 59 with low to moderate functionality). Children from families earning below 10,000€ are 2.58 times more likely to have low or moderate functionality compared to those from families earning between 10,001-20,000€ (OR = 2.58, 95% CI [1.21, 5.50]), highlighting the significant impact of lower socioeconomic status on child development. In this study, out of 44 children from families earning below 10,000€, 26 (or 59%) have severe autism (low or moderate functionality).
- Second-born children are 2.71 times more likely to show low or moderate functionality compared to first-born children (OR = 2.71, 95% CI [1.70, 4.29]). Specifically, we found that among first-born children, about 29.6% (103 out of 348) have severe autism, while among second-born children, about 56.9% (78 out of 137) have severe autism. This indicates that birth order has a notable effect on the child's developmental outcomes.
- Children whose mothers have a family history of autism, developmental disorders, epileptic seizures, depression, or anxiety are 42.5% more likely to have low or moderate functionality (severe autism) compared to high functionality (mild autism) (OR = 0.58, 95% CI [0.36, 0.91]). This suggests that such a family history may reduce the chances of higher functionality in children with ASD. However, in the univariate analysis, we found that children with a family history are more likely to have mild autism (73.05%) compared to children without a family history (57.14%). This seems to contradict the regression findings, which account for multiple factors.
- Mothers who experienced a viral or bacterial infection during pregnancy were 51% less likely to have children with high functionality (OR = 0.49, 95% CI [0.25, 0.97]). We found that 26.6% (17 out of 64) of children whose mothers had an infection exhibit severe autism (low to moderate functionality). This underscores the potential negative impact of maternal infections on the child's development. Vaginal bleeding during pregnancy posed significant risks: mothers who experienced bleeding during the first trimester were 6.43 times more likely to have children with low or moderate functionality (OR = 6.43, 95% CI [1.55, 26.77]), while third-trimester bleeding increased this likelihood by 13 times (OR = 13.02, 95% CI [2.15, 78.93]). We found that 46.3% of children whose mothers experienced vaginal bleeding during pregnancy show severe autism, compared to 35.4% for mothers who did not experience bleeding, while 68.42% (13 out of 19) of ASD children with 3rd trimester bleeding have severe autism. These findings emphasize the potential risks that pregnancy complications can pose to child development.
- Finally, babies who cried immediately after birth were 61% less likely to have low or moderate functionality (OR = 0.39, 95% CI [0.19, 0.81]). Among babies who cried immediately after birth, we found that 33.96% (144 out of 424) have severe autism. This early health indicator suggests a strong positive association with better functionality outcomes.”
DISCUSSION
384-392
“Prior research [33] has shown that older parental age at the time of birth is correlated with both the severity and 385 incidence of ASD in their progeny. Still, a substantial disparity was discovered in the age 386 of diagnosis, where children with greater functionality were diagnosed at a later age than those with lower functional capacity. For every additional year's postponement in the diagnoses of ASD, the likelihood of the child having low or moderate functionality diminishes by 35%. This suggests that an earlier diagnosis is associated with higher functionality in children with ASD and may be related to more pronounced ASD symptoms, as the functioning of children may be enhanced with earlier interventions [34,35].”
Revise this period, please. I did not detect a discussion of your results (Parental age and age of diagnosis). Also, you should connect these results with other studies investigating fathers’ age.
Thank you for your valuable feedback and for highlighting the potential discussion point regarding the impact of fathers’ age on the age of diagnosis of ASD in children. We appreciate the depth of review and the suggestion to connect our findings with other studies investigating paternal age.
In our study, the questionnaires were specifically administered to mothers, who were the primary caregivers, and thus, our data collection focused primarily on maternal insights and reported information. Due to this methodological framework, our analysis was oriented towards understanding factors from the perspective of the mothers, and we did not collect detailed demographic or diagnostic information directly from fathers.
Given this context, our study was not equipped to directly investigate the implications of fathers' age, which is why this aspect was not discussed in depth within our results. While we recognize the importance and relevance of paternal age in ASD research, as evidenced by existing literature, our current study scope and data limitations restricted our ability to engage with this variable comprehensively.
Check “earlier diagnosis is associated with higher functionality”.
Earlier diagnosis is linked with more severe autism, while later diagnosis is associated with milder forms of autism.
Check “543% higher odds” .. 1202% higher odds.
This was deleted
403
Similarly to the previous section, you could discuss these results in detail.
Answer to the reviewer
Thank you for your suggestion to further enrich the discussion of our results. We appreciate your attention to ensuring that our findings are thoroughly analyzed and contextualized within the broader research field.
We have carefully considered the balance between detailed discussion and maintaining readability and focus throughout our manuscript. The current structure and content of the discussion section have been deliberately crafted to align closely with our study's specific objectives and research questions. We aimed to provide a comprehensive yet concise analysis that directly addresses our findings while avoiding an overly extensive discussion that might dilute the focus or overwhelm the reader.
We believe that adding more detailed discussion could potentially obscure the key messages and findings of our study, making it harder for readers to extract the essential insights. However, we have ensured that critical points are discussed in depth, with references to relevant studies and theoretical implications, to provide a robust understanding of our results within the existing literature.
If there are specific areas within the results that you believe require further expansion, we would be grateful for your guidance on these points.
Thank you once again for your constructive feedback.
429
Low income could be caused by having a child with a severe diagnosis and not conversely. For Example, mothers often leave their jobs to assist their child decreasing their income. Revise with caution this section, please.
ANSWER TO THE REVIEWER
Thank you for your insightful comment regarding the interpretation of the relationship between low income and having a child with a severe diagnosis in our study. We appreciate your recommendation to approach this discussion with caution and to consider the directionality of this relationship.
In light of your feedback, we revised this section ( lines 590-599) to more accurately reflect the complexity and potential bi-directionality of the association between low income and the severity of a child's diagnosis. It is indeed possible, as you pointed out, that the financial burdens associated with caring for a child with severe conditions could lead to a reduction in family income, particularly if a parent, often the mother, reduces working hours or leaves employment altogether to provide necessary care.
Check 171% heightened likelihood.
This was deleted
Concluding.
The manuscript is clear and informative even if it lacks information regarding methodologies (sampling, instruments, variables, and reliability). The authors describe more tables than discuss results.
The text is redundant in diverse parts without addressing, in detail, the methodology and research process.
Since the research provides a survey its content should be described with caution, mere correlations could be biased, overall if the groups were not balanced.
Moreover, mothers with a child with a greater severity could report a more hard retrospective perception. The authors do not consider this typology of bias during the interview. Similarly, people with low-functioning children need more assistance, as a result, mothers could leave their jobs to give full assistance to children influencing their income. The regression model could not explain the entire phenomenon.
In the discussion section, you could emphasize your data and connection with previous literature, overall direct measurements, and longitudinal studies, necessary to respond to these research questions.
Your research after major revision merits consideration for publication.
I hope my recommendations will increase the quality of your research.
Good luck with your manuscript.
Overall Comments to the Reviewer
Thank you for your detailed review and constructive suggestions regarding our manuscript. We appreciate the time you took to evaluate our work and your recommendations, which we believe will greatly enhance the quality and clarity of our research. We are committed to addressing each point you raised to ensure our manuscript meets the rigorous standards of your journal.
- Methodology Clarifications: We acknowledge your observation that our manuscript could benefit from a more detailed description of the methodologies used, particularly regarding sampling, instruments, variables, and reliability. We revised the Method section to include more specific information about these aspects, ensuring a clear understanding of how data was collected, the tools used, and the reliability of these instruments.
- Discussion of Results vs. Description of Tables: We appreciate your point regarding the balance between describing tables and discussing results. In response, we reduced the redundancy in table descriptions and focus more on interpreting the results, linking them effectively to the research questions and objectives outlined.
- Addressing Potential Biases: Your comments about the potential biases due to retrospective perception by mothers and the economic impact of caregiving on income are extremely valuable. We enhanced our Discussion section to include a critical examination of these biases. This involves a discussion on how such factors might influence the reported data and the implications this has for interpreting our findings.
- Statistical Modeling and Interpretation: We agree that our regression model, while insightful, does not capture the entirety of the complex phenomenon of how having a child with ASD impacts family dynamics and socioeconomic status. We revised our analysis section to acknowledge these limitations more explicitly and discuss the potential for alternative models or methodologies that could provide deeper insights.
- Linking to Previous Literature and Emphasizing the Need for Longitudinal Studies: In the revised Discussion section, we tried to strengthen the connections to previous literature, emphasizing how our findings align with or differ from existing research. We also discuss the necessity of direct measurements and longitudinal studies to further validate and build upon our findings, as suggested.
We are optimistic that these major revisions will significantly elevate the quality of our manuscript and ensure its contribution is both clear and valuable to the field. Thank you once again for your thoughtful recommendations and encouragement. We look forward to your consideration of our revised manuscript
Round 2
Reviewer 1 Report
Comments and Suggestions for Authors
The authors integrated the recommendations from the previous review, resulting in significant improvements to the manuscript.
Based on these revisions, publication I recommend the manuscript for publication.
Author Response
Dear Reviewer,
Thank you very much for your positive feedback and recommendation for publication. We truly appreciate your constructive comments throughout the review process, which have significantly strengthened our manuscript. We are grateful for your support and consideration.
Kind regards,
A. Sousamli
Reviewer 2 Report
Comments and Suggestions for Authors
Minor concerns:
102-106
"Specifically, the objectives include (1) quantifying the influence of prenatal exposures to environmental toxins on the risk of ASD diagnosis, (2) assessing the impact of familial genetic history on ASD incidence, and (3) evaluating how these factors vary regionally across Greece. This analysis aims to contribute to targeted public health strategies and inform policy development for early ASD detection and intervention".
I don't suggest numbering the purpose of your study than research questions since you could need to recall them in the discussion section.
120-149
author better specified the method followed.
127
"Autism Spectrum Disorder (ASD)", only ASD throughout the manuscript.
134-136
"The sampling strategy was designed to reflect the demographic distribution of the Greek population. The sample was stratified based on key geographic and demographic variables to ensure comprehensive representation".
Thank you for clarifying the source of your sampling process so you can add the percentage of proportions of various regions.
For example, 500 mothers (20% first region, 30% second)
Also, 20 % from the community center and 25% from health services.
In that manner, your sampling may be more replicable.
142
"Additional stratification was applied based on socioeconomic status, wherein participants were grouped into quartiles based on the income distribution data available from national statistics".
I suggest adding a range of incomes in brackets showing quartiles or a website concerning national data with accessed dates.
150
"A structured questionnaire was developed to gather data, incorporating closed-ended and open-ended questions."
In order to increase the impact of your paper, I highly recommend adding (Appendix A) to the questionnaire and related responses.
156
Check this section since it is redundant from the previous one.
222-227
Check this section since it is redundant from the previous one.
228
The data analysis section is detailed, showing independent and dependent variables.
Results section
The authors have added the clinical description as suggested.
However, the graphs have low readability. Pay attention to adding notes under graphs and descriptions in the text, avoiding redundancies.
The authors reported some interpretations of their results.
In my opinion, you can increase the readability of your manuscript by working on the writing, removing redundancies, and completing some sections. For example, in the method section, you describe several times the sample. The addition of the questionnaire will clarify the comprehension of your variables. Likewise, since your data are not always coherent with the literature, you should describe them referring to methodological biases such as a partial sample (only mothers and limited numerosity) with few interactions (input variables) and no medical/genetic/psychiatric assessment.
Finally, I have had doubts regarding the title "Unveiling Hidden Risks: Prenatal and Demographic Insights into Autism Spectrum Disorder" since the keywords of your survey are not listed, limiting the search of your paper from other researchers.
Consider please only the following example to explain what I intend:
Pregnancy and post-natal risk factors related to future autism diagnosis, and symptoms' severity: a national survey.
I hope my suggestions will help your editorial process.
Good luck
Author Response
Minor concerns:
REVIEWER’S COMMENT
102-106
"Specifically, the objectives include (1) quantifying the influence of prenatal exposures to environmental toxins on the risk of ASD diagnosis, (2) assessing the impact of familial genetic history on ASD incidence, and (3) evaluating how these factors vary regionally across Greece. This analysis aims to contribute to targeted public health strategies and inform policy development for early ASD detection and intervention".
I don't suggest numbering the purpose of your study than research questions since you could need to recall them in the discussion section.
Reply to the reviewer: Thank you for your suggestion. We have removed the numbering of the study's purpose and research questions to ensure smoother integration in the discussion section. The revisions have been made accordingly.
REVIEWER’S COMMENT
120-149
author better specified the method followed.
Reply to the reviewer: Thank you for your comment. We revised the methodology accordingly.
REVIEWER’S COMMENT
127
"Autism Spectrum Disorder (ASD)", only ASD throughout the manuscript.
Reply to the reviewer: Thank you for your comment. We use only ASD throughout the manuscript as you pointed out.
REVIEWER’S COMMENT
134-136
"The sampling strategy was designed to reflect the demographic distribution of the Greek population. The sample was stratified based on key geographic and demographic variables to ensure comprehensive representation".
Thank you for clarifying the source of your sampling process so you can add the percentage of proportions of various regions.
For example, 500 mothers (20% first region, 30% second)
Also, 20 % from the community center and 25% from health services.
In that manner, your sampling may be more replicable.
Reply to the reviewer: Thank you for your insightful suggestion. We have added the percentage breakdown of the sample drawn from the various regions and service types. We added: “The study's sample was proportionally stratified to ensure a representative distribution across the country. Specifically, of the 517 mothers included, 20% were from Central Macedonia and Thrace, 30% from Epirus and Thessaly, 15% from Crete, and the remaining 35% were distributed across Attica, Peloponnese, and the Aegean islands. Furthermore, 40% of the sample was recruited from Specialized Educational Institutions (e.g., special needs schools and autism-specific day schools), 30% from Therapeutic Day Centers, 20% from Community-based Support Centers, and the remaining 10% were recruited from Pediatric and Developmental Health Clinics.”
REVIEWER’S COMMENT
142
"Additional stratification was applied based on socioeconomic status, wherein participants were grouped into quartiles based on the income distribution data available from national statistics".
I suggest adding a range of incomes in brackets showing quartiles or a website concerning national data with accessed dates.
Reply to the reviewer: Thank you for your suggestion. We added the following paragraph “Additional stratification was applied based on socioeconomic status, wherein participants were grouped into quartiles according to the income distribution data available from national statistics. The income quartiles were delineated as follows: Quartile 1: less than €10,000, Quartile 2: €10,001 to €20,000, Quartile 3: €20,001 to €40,000, and Quartile 4: more than €40,001. These ranges reflect the national income distribution as per the most recent data available from the Hellenic Statistical Authority (accessed on 12th October 2024) which can be found at: [https://www.statistics.gr/documents/20181/262f2183-1930-6ece-a792-e02103dfbe00]”
REVIEWER’S COMMENT
150
"A structured questionnaire was developed to gather data, incorporating closed-ended and open-ended questions."
In order to increase the impact of your paper, I highly recommend adding (Appendix A) to the questionnaire and related responses.
Reply to the reviewer Thank you for your suggestion. We have now included the questionnaire in Appendix A to enhance the impact of the paper.
REVIEWER’S COMMENT
156
Check this section since it is redundant from the previous one.
Reply to the reviewer Checked!
REVIEWER’S COMMENT
222-227
Check this section since it is redundant from the previous one.
Reply to the reviewer Checked!
REVIEWER’S COMMENT
228
The data analysis section is detailed, showing independent and dependent variables.
Results section
The authors have added the clinical description as suggested.
However, the graphs have low readability. Pay attention to adding notes under graphs and descriptions in the text, avoiding redundancies.
Reply to the reviewer Thank you for your feedback. We have addressed the issue by adding explanatory notes under each graph to enhance readability and provide clear context. Additionally, we have revised the text to avoid redundancies and ensure smoother integration of the graphs with the descriptions.
REVIEWER’S COMMENT
The authors reported some interpretations of their results.
In my opinion, you can increase the readability of your manuscript by working on the writing, removing redundancies, and completing some sections. For example, in the method section, you describe several times the sample. The addition of the questionnaire will clarify the comprehension of your variables. Likewise, since your data are not always coherent with the literature, you should describe them referring to methodological biases such as a partial sample (only mothers and limited numerosity) with few interactions (input variables) and no medical/genetic/psychiatric assessment.
Reply to the reviewer Thank you for your insightful suggestions. We have revised the manuscript to improve its readability by removing redundancies, particularly in the method section where the sample was described multiple times. We have also included the questionnaire to clarify the comprehension of our variables.
Additionally, we have enhanced the discussion section by addressing potential methodological biases, such as the partial sample (limited to mothers), the relatively small sample size, limited interaction variables, and the absence of medical, genetic, and psychiatric assessments. These limitations are now clearly acknowledged to explain any discrepancies between our findings and existing literature. We appreciate your suggestions and believe these changes improve the clarity and impact of the manuscript.
REVIEWER’S COMMENT
Finally, I have had doubts regarding the title "Unveiling Hidden Risks: Prenatal and Demographic Insights into Autism Spectrum Disorder" since the keywords of your survey are not listed, limiting the search of your paper from other researchers.
Consider please only the following example to explain what I intend:
Pregnancy and post-natal risk factors related to future autism diagnosis, and symptoms' severity: a national survey.
I hope my suggestions will help your editorial process.
Good luck
Reply to the reviewer: Thank you for your suggestion regarding the title. We agree that including specific keywords could improve the searchability of the paper. We have revised the title to better reflect the key terms of the study while maintaining an academic tone. The new title is: Perinatal and Demographic Risk Factors Associated with Autism Spectrum Disorder: A National Survey of Potential Predictors and Severity
Best Regards,
A. Sousamli